# A macroecological perspective on genetic diversity in the human gut microbiome

**William R. Shoemaker**◯¤*

Department of Ecology and Evolutionary Biology, University of California, Los Angeles, Los Angeles, California, United States of America

¤ Current address: The Abdus Salam International Centre for Theoretical Physics (ICTP), Trieste, Italy
* williamrshoemaker@gmail.com

## Abstract

While the human gut microbiome has been intensely studied, we have yet to obtain a sufficient understanding of the genetic diversity that it harbors. Research efforts have demonstrated that a considerable fraction of within-host genetic variation in the human gut is driven by the ecological dynamics of co-occurring strains belonging to the same species, suggesting that an ecological lens may provide insight into empirical patterns of genetic diversity. Indeed, an ecological model of self-limiting growth and environmental noise known as the Stochastic Logistic Model (SLM) was recently shown to successfully predict the temporal dynamics of strains within a single human host. However, its ability to predict patterns of genetic diversity across human hosts has yet to be tested. In this manuscript I determine whether the predictions of the SLM explain patterns of genetic diversity across unrelated human hosts for 22 common microbial species. Specifically, the stationary distribution of the SLM explains the distribution of allele frequencies across hosts and predicts the fraction of hosts harboring a given allele (i.e., prevalence) for a considerable fraction of sites. The accuracy of the SLM was correlated with independent estimates of strain structure, suggesting that patterns of genetic diversity in the gut microbiome follow statistically similar forms across human hosts due to the existence of strain-level ecology.

## Introduction

The human gut microbiome harbors astounding levels of genetic diversity. Hundreds to thousands of species continually reproduce in a typical host, accruing a total of $\sim 10^9$ *de novo* mutations each day [1]. Due to the comparatively brief generation time of microbes in the human gut [2], those mutations that are beneficial can rapidly fix on a timescale of days to months [1, 3–9]. Such evolutionary dynamics have the capacity to alter the genetic composition of a species within a given host. However, while all genetic diversity ultimately arises due to mutation, this actuality does not mean that all the genetic variants observed in the human gut are necessarily subject to evolutionary dynamics.

For many bacterial species a large number of genetic variants do not fix or become extinct within a given host. Instead, these variants fluctuate at intermediate frequencies over time on timescales ranging from months to years [1, 10–13]. Such within-host genetic structure is

**Data Availability Statement:** All raw sequencing data for the metagenomic samples used in this study were downloaded from the HMP \cite{huttenhower_structure_2012, lloyd-price_strains_2017} (URLs: \url{https://portal.

hmpdacc.org/}, \url{https://aws.amazon.com/datasets/human-microbiome-project/}). \texttt{MIDAS} output was processed using publicly available code \cite{garud_evolutionary_2019}. Processed data is available on Zenodo: \url{https://doi.org/10.5281/zenodo.6793770}. All code written for this study is available on GitHub: \url{https://github.com/wrshoemaker/StrainPrevalence}.

**Funding:** This work was supported by the NSF Postdoctoral Research Fellowships in Biology Program under Grant No. 2010885 (W.R.S.). https://beta.nsf.gov/funding/opportunities/postdoctoral-research-fellowships-biology-prfb The funders had no role in study design, data collection and analysis, decision to publish, or preparation of the manuscript.

**Competing interests:** The authors have declared that no competing interests exist.

reflected by the shape of phylogenetic trees constructed from microbial isolates, where the existence of a low number of deep phylogenetic branches suggests the existence of strain structure [11, 14–19]. This pattern of diversity within hosts arises due to the co-occurrence of a few ($\sim \mathcal{O}(1-4)$) genetically and ecologically diverged strains that belong to the same species, a process known as oligocolonization [3, 11]. This sub-species ecological structure that can occur within a host is more than a descriptive detail, as it has been proposed that strains are the relevant scale at which interactions and dynamics occur in microbial systems [20, 21]. Thus, the dynamics of the genetic variants that comprise a given strain are subject to exogenous and endogenous ecological processes [22–24]. However, evolution within a strain does not stop, as genetic variants continue to be acquired and segregate over time within a given strain [5]. Such dynamics are a clear departure from those captured by standard population genetic models used to describe microbial evolution, where genetic variants either arise in a population due to mutation or are introduced by migration and then proceed towards extinction or fixation (i.e., origin-fixation models), suggesting that measures of genetic diversity estimated within the human gut are shaped by the ecology of strains alongside evolutionary processes such as low recombination rates that result in physical linkage between alleles [3, 5, 10, 25].

This confluence of ecological and evolutionary dynamics requires new approaches and theory for characterizing genetic diversity in the human gut. Many studies tackle such complexity by examining individual species [6, 26–28] or by searching for genetic differences between species [11, 12, 29–33]. While such approaches are useful for identifying individual species that are potential contributors towards specific conditions such as disease or the ability to metabolize certain resources, it is difficult to translate isolated observations into general patterns. By focusing on individual species and differences between species it is plausible that uncharacterized patterns of genetic diversity that are generalizable across species may have been overlooked.

As an alternative, it is reasonable to first identify genetic patterns that are similar across species (i.e., statistical invariance). Such an approach may provide the empirical motivation necessary to identify mathematical models that can explain said patterns and aid in the identification of underlying ecological or evolutionary dynamics [10, 34–36]. In recent years, substantial progress has been made towards characterizing the typical microbial evolutionary dynamics across species that operate within and across human hosts [3, 4, 7, 8, 37–39]. An example of such a pattern is the observation that the relationship between synonymous nucleotide divergence (a proxy for evolutionary time) and the ratio of nonsynonymous and synonymous divergence ($dS$ vs. $dN/dS$) falls on a single curve across microbial species in the human gut, representing 20 genera, 14 families, 7 orders, 6 classes, and 5 phyla [3, 39]. While this approach can often be limited by the number of observations and measurement error, modern data curation methods can often alleviate these limitations. Using this approach it is possible to leverage the richness of species in the human gut microbiome, where each species can be viewed as a draw from an unknown distribution and, as an ensemble, be used to identify patterns that are statistically invariant [40, 41].

To identify such patterns, it is useful to examine prior attempts that successfully pared down the complexity of the gut. Notable recent examples come from the discipline of macroecology, which has succeeded at predicting patterns of microbial diversity and abundance at the species level across disparate environments, including the gut microbiome [42–47]. This approach emphasizes the benefits of identifying patterns of diversity that are statistically invariant, motivating the development of quantitative predictions derived from ecological first principles. Recent work suggests that this approach may hold across scales of organization in the human gut, as species-level macroecological patterns have been extended to temporal

patterns of strain-level ecology within a single host [46]. This consistency in strain-level patterns provided the motivation to apply an established model of ecology to predict macroecological quantities within a single host over time, the Stochastic Logistic Model of growth (SLM). In macroecology, the SLM has been found to successfully characterize the distribution of species relative abundance across hosts and over time within a host (i.e., the Abundance Fluctuation Distribution (AFD)), the relationship between the mean abundance of a species and the fraction of hosts where it is present (i.e., the abundance-prevalence relationship [48]), and the relationship between the mean and variance of the abundance of a species (i.e., Taylor's Law [49]) [44]. Inspired by the success of the SLM, it was recently applied to the strain-level to explain the temporal form of the AFD and Taylor's Law within a single human host [46]. In this study it was found that the temporal dynamics of strains within a single healthy human host invariant with respect to time (i.e., stationary). Motivated by this result, it was determined that the empirical distribution of strain frequencies over time followed the distribution of the SLM at stationarity, a gamma distribution. The results of this study suggest that the SLM, a model that succeeded in predicting patterns of strains within a single human host, may also succeed in predicting patterns of genetic diversity across unrelated hosts due to the existence of strain structure.

In this study, I sought to determine whether the SLM as a model of ecology was capable of quantitatively predicting patterns of genetic diversity across hosts due to the existence of strain structure. I identified patterns of diversity that remained statistically invariant among phylogenetically distant species, providing the motivation necessary to identify the SLM as a plausible model of across-host patterns of diversity. To evaluate the feasibility of the SLM while accounting for the effects of sampling, I obtained predictions for the fraction of hosts harboring an allele at a given nucleotide site (i.e., prevalence) using zero free parameters. I identified evolutionary models of allele frequencies that predict the same stationary probability distribution as the SLM and found that their assumptions are unrealistic to explain the data. To confirm that the success of the SLM was due to the presence of strains, I inferred whether strain structure was present in each host for each species using an established computational approach, finding that the presence of strain structure was correlated with the accuracy of the SLM in predicting allelic prevalence.

## Results

### Patterns of genetic diversity are statistically invariant across species

In order to determine whether it is possible to predict patterns of genetic diversity in the human gut, it is necessary to first investigate the degree of similarity in measures of genetic diversity across phylogenetically distant species. This manner of visualization, known as a data collapse, allows one to assess whether it is reasonable to assume that similar dynamics underlie different systems [50–52]. Such an analysis also provides the benefit of allowing for the identification of previously unknown empirical patterns for subsequent investigation. To determine whether there is evidence that the distributions of measures of genetic diversity have qualitative similar forms across species, I compiled allele frequency data for 22 bacterial species across human hosts using a quality control pipeline that explicitly accounted for the rate of sequencing error using the Maximum-likelihood Analysis of Population Genomic Data `MAPGD` program (Materials and methods). The total number of processed hosts ranged from 108–371 across species, with a median of 182 (Fig 1a). The total number of sites ranged from 39–37, 204 across species, with a median of 10,269 synonymous and 5,204 nonsynonymous sites (Fig 1b, S1b Fig). These results, and their existence for both synonymous and nonsynonymous sites, provides the empirical basis necessary to formulate quantitative predictions.

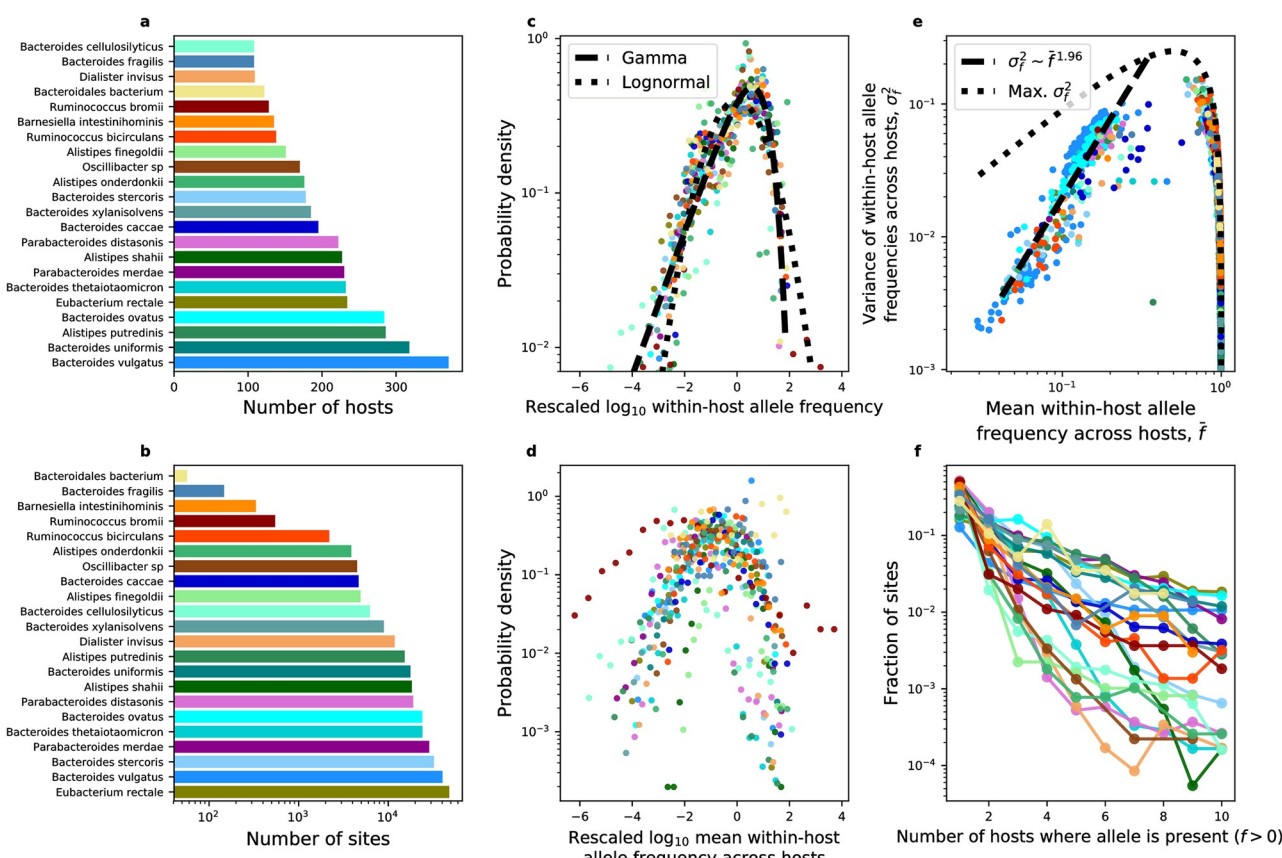

**Fig 1. Distributions of genetic diversity exhibit similar statistical forms across phylogenetically distant species in the human gut. a,b)** Similarity in patterns of genetic diversity was evaluated for sites obtained from the 22 most prevalent bacterial species. **c)** The distribution of within-host allele frequencies across all hosts as well as **d)** the distribution of mean allele frequencies were rescaled to determine whether they exhibited similar forms, specifically by rescaling their logarithm using the standard score (i.e., z-score). In **c**, statistical fits of a gamma (the distribution predicted by the SLM) and a lognormal (a point of comparison) are illustrated as black lines. To limit the effect of the bounded nature of allele frequencies on the distribution, mean frequencies containing observations of $f = 1$ were excluded from subplot **d**. **e)** The relationship between statistical moments of within-host allele frequencies was consistent across species, as there was a strong linear relationship between the mean frequency of an allele and its variance on a log-log scale (i.e., Taylor's Law). To reduce the contribution of an excess number of zeros towards estimates of $\bar{f}$ and $\sigma_f^2$, alleles with non-zero values of $f$ in <35% of hosts were excluded. **f)** Finally, the fraction of sites harboring alleles present in a given number of hosts decreased in a similar manner across species. All sites in this analysis are synonymous, identical analyses were performed on alleles at nonsynonymous sites (S1 Fig). Species within the same genus were assigned the same primary or secondary color with different degrees of saturation.

First, I obtained the distribution of across-host allele frequencies for each nucleotide site and then pooled the frequencies of all sites. If the typical allele was present due to evolutionary processes, then the empirical distribution of within-host allele frequencies across hosts would be the equivalent to the ensemble of single-site frequency spectra expected from within-host evolution [53, 54]. If the typical allele was present because it was on the background of a strain, then the macroecological view of this distribution is that it captures the distribution of relative abundances of strains across hosts, the AFD [44]. Furthermore, the degree of similarity across species allows one to assess whether it is reasonable to predict that a single probability distribution is capable of explaining the distribution of within-host allele frequencies across hosts for phylogenetically distant species.

In order to determine whether different distributions share a single form it is useful to rescale them by key parameters [50]. Inspired by prior work [44], I rescaled the distribution of within-host frequencies across hosts by 1) pooling all non-zero frequencies for a given species,

2) log-transforming all frequencies, 3) calculating the mean and standard deviation of the frequency, and 4) calculating each log-transformed frequency as a standard score (i.e., z-score). By repeating this process for each species, one can determine whether the form of the distribution qualitatively varies across species or whether they simply differ in their statistical moments. The distribution of within-host allele frequencies had a similar qualitative form across species in the human gut for synonymous (Fig 1c) and nonsynonymous sites (S1c Fig), suggesting that a single distribution may be sufficient to characterize all species. Regions of the distribution are well-captured by the gamma distribution [44], suggesting that it would be informative to examine models that lead to a gamma distribution and then assess the gamma's capacity to predict quantities calculated from individual alleles. As a point of comparison, I fit the distribution in Fig 1c using a lognormal distribution. This distribution was previously used to evaluate the AFD at the species level in disparate ecosystems [44]. The lognormal clearly deviates from the bulk of the distribution, a result that is even more apparent when the probability density is plotted as a survival probability (S2a and S3a Figs). An Akaike Information Criterion (AIC) test supports this conclusion (Synonymous: $AIC_{gamma}$ = 6, 277, 330, $AIC_{lognormal}$ = 6, 618, 492; Nonsynonymous: $AIC_{gamma}$ = 3, 359, 295, $AIC_{gamma}$ = 3, 490, 025).

Beyond the shape of the distribution of within-host allele frequencies, the statistical moments of within-host frequencies across hosts also exhibit qualitatively similar forms. I repeated the same standard score rescaling procedure for the logarithm of the mean within-host allele frequency across hosts ($\overline{f}$). The distribution of $\overline{f}$ tended to overlap across species for both synonymous (Fig 1d, S2b Fig) and nonsynonymous sites (S1d and S3b Figs). While here I am not explicitly interested in the processes that shape the mean distribution, as the mean will be used as an empirical input for calculating predictions in the subsequent section, the result does suggest that statistical moments calculated across host display features of invariance.

The mean and variance of random variables frequently follows linear relationships on logarithmic scales across biological systems, most notably in patterns of biodiversity in ecological communities [44, 55, 56] but also among population genetic patterns [57–59]. The existence of this relationship, known as Taylor's Law [49], would imply in the context of this study that the mean and variance of allele frequencies across hosts are not independent among species, reducing the number of parameters necessary to characterize the dynamics of the system [44]. Furthermore, if the variance scales quadratically with the mean then the existence of the relationship implies that the coefficient of variation of $f$ is constant across sites, an observation that can considerably reduce the difficulty of characterizing the dynamics of the system.

By examining the relationship between $\overline{f}$ and the variance of $f$ ($\sigma_f^2$), it is clear that the two moments follow a linear relationship on a logarithmic scale for low values of $\overline{f}$ ($0 < \overline{f} \lesssim 0.35$). The exponent of this relationship is $\sim 1.96$ (bootstrapped 95% CI from 10,000 samples: [1.85, 2.06]) for synonymous sites, a value that is remarkably close to two, implying that the coefficient of variation in $f$ can be viewed as a constant for the range of $\overline{f}$ where the relationship is linear (Fig 1e). The exponent is slightly reduced for nonsynonymous sites ($\sim 1.83$, 95% CI [1.72, 1.93]; S1e Fig), suggesting that the variance increases with the mean at a slower rate relative to synonymous sites. Given that purifying selection is pervasive across species within the human gut [3, 12, 39, 60], it is likely that the typical allele at a nonsynonymous site confers a deleterious fitness effect, reducing its variance across hosts for low values of $\overline{f}$. However, the linear relationship does not extend to high values of $\overline{f}$. Given that $f$ is, by definition, a bounded quantity (i.e., $0 \leq f \leq 1$), it is possible that the relationship between $\overline{f}$ and $\sigma_f^2$ for values of $\overline{f} \approx 1$ is governed by the upper bound on $f$. To determine whether this is the case, I plotted the maximum possible value of $\sigma_f^2$ for a given value of $\overline{f}$ constrained on the lower and upper

bounds of $f$. This relationship, known as the Bhatia–Davis inequality, is defined as $\sigma_f^2 \leq (\max(f) - \overline{f})(\overline{f} - \min(f)) = (1 - \overline{f})\overline{f}$ [61]. The empirical relationship between $\overline{f}$ and $\sigma_f^2$ follows the inequality across species for $\overline{f} \gtrsim 0.35$, suggesting that the relationship can be explained solely by the mathematical constraints on $f$, making the relationship uninformative for the purpose of identifying universal evolutionary or ecological patterns at a certain scale. It is worth noting that an exponent of two can emerge if the underlying distribution is sufficiently skewed [62, 63]. However, similar to ecological analyses of the relationship between the mean and variance of species abundances [44, 56, 64], the fact that the observed mean allele frequency varies by close to two orders of magnitude suggests that this patterns reflects a true scaling relationship. While the existence of this relationship may not be system-specific [64], it does allow us to make a claim about the relationship between statistical moments.

Finally, I turned my attention to the fraction of hosts where a given allele is present. I found that the number of hosts in which a typical allele is present is small for synonymous (Fig 1f) and nonsynonymous sites (S1f Fig). Alternatively stated, the fraction of hosts harboring a given allele (i.e., prevalence) is typically low.

## Predicting the prevalence of an allele across hosts

The existence of multiple patterns of genetic diversity that are universal across evolutionarily distant bacterial species suggests that comparable dynamics are ultimately responsible. The next task is to identify a candidate model capable of explaining said patterns. The shape of the rescaled distribution of allele frequencies suggests that a gamma distribution is a suitable candidate, reducing the range of feasible models to those that are capable of predicting said distribution or a distribution of similar form. Different approaches can be used to identify such a model. However, given the consistency of the patterns, it is appropriate to focus on models that solely contain parameters that can be measured from empirical data (i.e., no statistical fitting) rather than relying on estimates of free parameters via statistical inference (i.e., statistical fitting) [65].

We begin with the assumption that the frequency dynamics of a typical allele within a host are primarily driven by the ecological dynamics of the strain on which said allele resides. Appropriate Langevin equations (i.e., stochastic differential equations) that capture relevant ecological dynamics can be used. However, regardless of the underlying dynamics, the available data constrains the ways in which a given model can be evaluated. Given that temporal metagenomic data for the human gut microbiome remains restricted to a small number of hosts, I focused on samples taken at a single timepoint across a large number of unrelated hosts. This detail means that time-dependent solutions of the probability distribution of $f$ cannot be empirically evaluated (i.e., $p(f, t)$), so I instead focused on stationary probability distributions and limiting cases where time-dependence is captured by parameters that can, in principle, be estimated (i.e., $p(f)$). This assumption of stationarity (i.e., time-invariance) is supported by previous research efforts that examined macroecological patterns that were stationary with respect to time at the strain level [46].

To model the dynamics of an allele that is on the genetic background a strain, it is necessary to identify essential features of growth. There are two main features that are necessary to consider the deterministic dynamics of strain dynamics: 1) that the rate of growth is often exponential when a species or strain is far from its carrying capacity and 2) that growth is self-limiting. There is also the need to consider stochasticity in growth that can be driven by environmental noise. To capture these features I examined a Langevin equation known as the Stochastic Logistic Model of growth (SLM), a model that has recently been shown to describe a range of macroecological patterns for microbial communities across disparate environments

[44, 66–68] as well as the temporal dynamics of strains within a single human host [46]. The SLM is defined as

$$\underbrace{\frac{\partial f}{\partial t} = \frac{f}{\tau_i}\left(1 - \frac{f}{K_i}\right)}_{\text{Self-limiting growth}} + \underbrace{\sqrt{\frac{\sigma_{\tau_i}}{\tau_i}} f \cdot \eta(t)}_{\text{Environmental noise}} \tag{1}$$

Here I am assuming that the allele I observed within a given host was present because it was on the background of a strain. This interpretation means that the fluctuations of the allele over time are due to the ecological fluctuations of the strain, where the terms $\frac{1}{\tau_i}$, $K_i$, and $\sigma_{\tau_i}$ respectively represent the intrinsic growth rate, the carrying capacity within a given species in terms of relative abundance ($0 \leq K_i \leq 1$), and the coefficient of variation of growth rate fluctuations of the $i$th strain.

Environmental noise is captured by the product of a linear frequency term (as opposed to demographic noise, which would be captured by the term $\sqrt{f}$), the compound parameter $\sqrt{\frac{\sigma_{\tau_i}}{\tau_i}}$, and a Brownian noise term $\eta(t)$ that introduces stochasticity into the equation. Using standard definitions of Langevin equations, the expected value of $\eta(t)$ is $\langle \eta(t) \rangle = 0$ [69]. The dependence of $\eta(t')$ at time $t'$ on an earlier time $\eta(t)$ is defined as $\langle \eta(t)\eta(t') \rangle = \delta(t - t')$ [69]. This standard definition means that if the noise term is shifted in time, then it has zero correlation with itself, otherwise it is identical to itself.

This definition of a Langevin equation is convenient in that it is possible to obtain a partial differential equation describing how the probability distribution of $f$ changes with time (i.e., the Fokker-Planck equation) [69]. Once this equation is obtained, the probability distribution of $f$ at stationarity (i.e., no dependence on time) can be obtained. One finds that the SLM predicts that the frequency of a given allele on the background of a strain follows a gamma distribution (additional detail provided in Materials and methods)

$$f_{\text{SLM}} \sim \text{Gamma}\left(\frac{2}{\sigma_{\tau_i}} - 1, \frac{2}{K_i \sigma_{\tau_i}}\right) \tag{2}$$

This distribution is fully characterized by the mean frequency and the squared inverse of the coefficient of variation across hosts ($\beta = \frac{\bar{f}^2}{\sigma_f^2}$) across hosts [44, 70]

$$f \sim \text{Gamma}\left(\beta, \frac{\beta}{\bar{f}}\right) \tag{3}$$

The similarity in the shape of the distribution of $\bar{f}$ across species and the relationship between $\bar{f}$ and $\sigma_f^2$ suggests that the mean and variance are appropriate quantities to evaluate the predictive capacity of each model (Fig 1d and 1e, S1d and S1e Fig). However, $\bar{f}$ and $\sigma_f^2$ can be interpreted as parameters of the SLM (i.e., empirical inputs), meaning that the SLM cannot be used to predict $\bar{f}$ and $\sigma_f^2$. To test the applicability of the SLM, I chose to examine the fraction of hosts harboring a given allele (i.e., prevalence), a quantity that has been used to examine microbial ecology and evolution across systems [44, 71], including the human gut microbiome [3, 7, 44].

In order to test prevalence predictions, it is necessary to account for the sampling effort at a given site in a given host (i.e., total depth of sequencing coverage). This can be accomplished by deriving the sampling distribution of the gamma, providing the probability of observing $A$

reads of a gamma distributed allele with $D$ coverage.

$$\Pr[A|D, \beta, \beta/\overline{f}] = \frac{\Gamma(\beta + A)}{A!\Gamma(\beta)} \left(\frac{\overline{f}D}{\beta + \overline{f}D}\right)^A \left(\frac{\beta}{\beta + \overline{f}D}\right)^\beta \tag{4}$$

A value $A = 0$ represents the absence of an allele, which can be used to define presence as the complement, providing a natural definition of the prevalence of an allele across hosts.

$$\langle \varrho \rangle = 1 - \frac{1}{M}\sum_{m=1}^{M} \Pr[0|D_m, \beta, \beta/\overline{f}] \tag{5}$$

where I have defined prevalence as the average of the probability of presence over $M$ hosts. While Eq 5 is correct, the pipeline used for sequence data enacted a cutoff for the total depth of coverage, resulting in a coverage cutoff for a given minor allele ($A_{\text{cutoff}} = 10$; S4 Fig). This cutoff truncates the sampling distribution of minor allele read counts, meaning that read counts for a given allele less than the specified cutoff are effectively observed as zeros. This inferential detail can be explicitly accounted for by summing over the probabilities of observing alternative allele read counts up to and excluding the cutoff

$$\langle \varrho \rangle = 1 - \frac{1}{M}\sum_{m=1}^{M}\sum_{A_i=0}^{A_{\text{cutoff}}-1} \Pr[A_i|D_m, \beta, \beta/\overline{f}] \tag{6}$$

The choice of prevalence also allows one to evaluate the extent that the gamma distribution can recapitulate empirical relationships between genetic quantities. One such relationship is that the prevalence of a species (equivalently known as *occupancy* in macroecology) should increase with its mean abundance across communities [72], a pattern that has been found to exhibit statistically similar forms at the species level across microbial systems [48, 73, 74] and can be quantitatively explained through the existence of macroecological laws [44].

## The Stochastic Logistic Model succeeds at predicting allelic prevalence

By examining the relationship between observed and predicted allelic prevalence, I found that the SLM generally succeeded in predicting this relationship for both synonymous and nonsynonymous sites using zero free parameters (Fig 2a, S5 and S6 Figs). Alternatively stated, predictions were obtained by computing the predicted prevalence using Eq 6 without the need to perform statistical fitting. The fraction of all sites with relative errors $\leq 0.1$ ($\leq 10\%$) ranged from 0.19–0.6 across species, suggesting that a considerable fraction of genetic variants within the human gut are driven by ecological dynamics (Fig 2b, S7 and S8 Figs). Furthermore, the SLM generally succeeded in recapitulating the relationship between $\overline{f}$ and prevalence, a strain-level analogue of abundance-prevalence relationships in macroecology (Fig 2b, S9 and S10 Figs). While the SLM on its own cannot be used to predict Taylor's Law since the mean and variance were used as empirical inputs, the evidence for the existence of Taylor's Law constrains the parameterization of the SLM and its subsequent interpretation [44, 75]. For the sites that the SLM is able to predict with a high degree of accuracy, the existence of Taylor's Law implies that $\beta$ is constant across sites ($\sigma_{\tau_i} = \sigma_\tau$; Fig 1e, S12 and S13 Figs). Thus, the function for the expected allele frequency reduces to the proportionality $\langle f \rangle \propto K_i$.

These results suggest that it is worth investigating the accuracy of the SLM across observed estimates of prevalence. Given that a considerable fraction of sites have prevalence values close to one (e.g., the dot in the top-right corner of Fig 2b), where the SLM has its highest degree of accuracy, it is necessary to remove these sites in order to examine the full distribution of

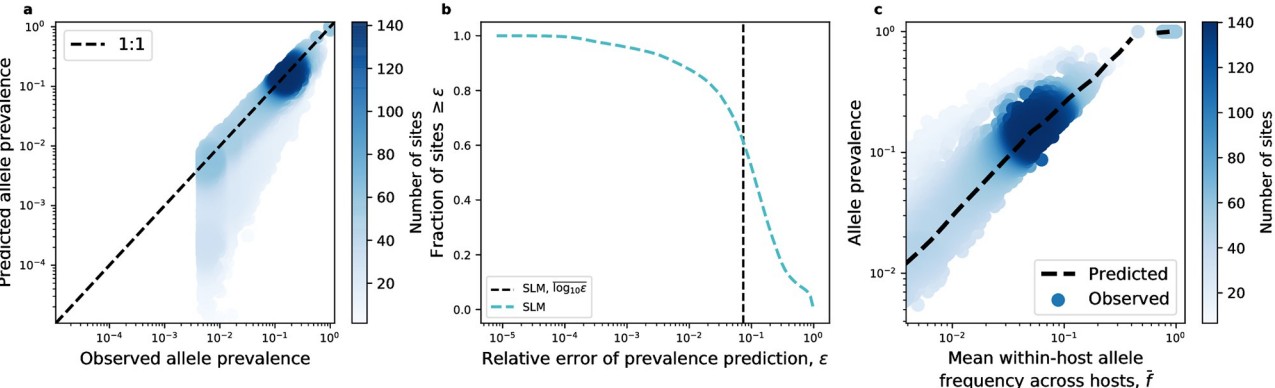

**Fig 2. The SLM successfully predicts genetic patterns for prevalent alleles. a)** Predicted values of prevalence were obtained (Eq 6) and matched with their observed values, where the data generally fell on the one-to-one line (dashed black line) for sites with a prevalence $\gtrsim 0.01$. **b)** The pattern of the SLM performing better for higher values of prevalence was illustrated by quantifying the relative error of the prevalence predictions. The mean of the logarithm of the relative error of the SLM over all sites ($\overline{\log_{10}\varepsilon}$, dashed black line) was $\sim 0.1$. **c)** The contingency of the SLM's success was illustrated by examining the relationship between the mean frequency of an allele across hosts ($\bar{f}$) and its prevalence. The predictions of the SLM (not a statistical fit) succeed for high mean frequency alleles (dashed black line). All analyses here were performed on alleles at synonymous sites using the common commensal gut species *B. vulgatus*. The color of each datapoint is proportionate to the number of sites. Visualizations of the predictions in this plot for all species for nonsynonymous and synonymous sites can be found in S1 Text.

prediction errors. By focusing on sites with observed prevalences <0.9, one can see that observed and predicted prevalence values followed a one-to-one relationship across a wide range of observed prevalence values for many species (Fig 3a). From these results, one can glean a few insights into the appropriateness of the SLM. First, for most observed prevalence values, when the predicted prevalence differs from the observed value it is generally below the one-to-one line. This pattern does not deviate as the observed prevalence increases, suggesting that the SLM is generally able to capture the distribution of *f* and its relation to prevalence for the range of observed prevalence values. When the SLM is inaccurate it generally underpredicts the true prevalence of an allele. Alternatively stated, the SLM can predict an excess of zero observations (*f* = 0). However, there is a large uptick in the predicted prevalence for alleles with high observed values of prevalence, where the predictions are effectively on the one-to-one line. This is the case, as there is a large drop in the error as observed prevalence increases (Fig 3b), suggesting a negative relationship between the observed prevalence of an allele and the relative error of the SLM. I found that the correlation between these two quantities was negative for all species (Fig 3c). By permuting the order of these quantities I obtained null distributions of correlation coefficients for all species, where the majority of coefficients (16/22) clearly fall below the bounds of their null 95% confidence intervals.

The dependence of the predictive success of the SLM on the observed prevalence of an allele across hosts is a curious pattern. Ideally, one expects that the SLM is capable of explaining the dynamics of alleles if they primarily exist on the genetic background of strains, as the SLM has been found to explain the temporal dynamics of strain frequencies within a single host [46]. The fact that the SLM, when it is inaccurate, tends to underpredict true prevalence allows one to rule out models that predict an excess proportion of values of *f* = 0 than expected from a given distribution (e.g., zero-inflated gamma), as they would further reduce the predicted prevalence and increase the error of the prediction. Such models are often representative of competitive exclusion [44], an ecological outcome where the presence of a given species precludes the existence of another species. In the context of this study, competitive exclusion corresponds to the hypothesis that a strain is not found in a given host because it is unable to

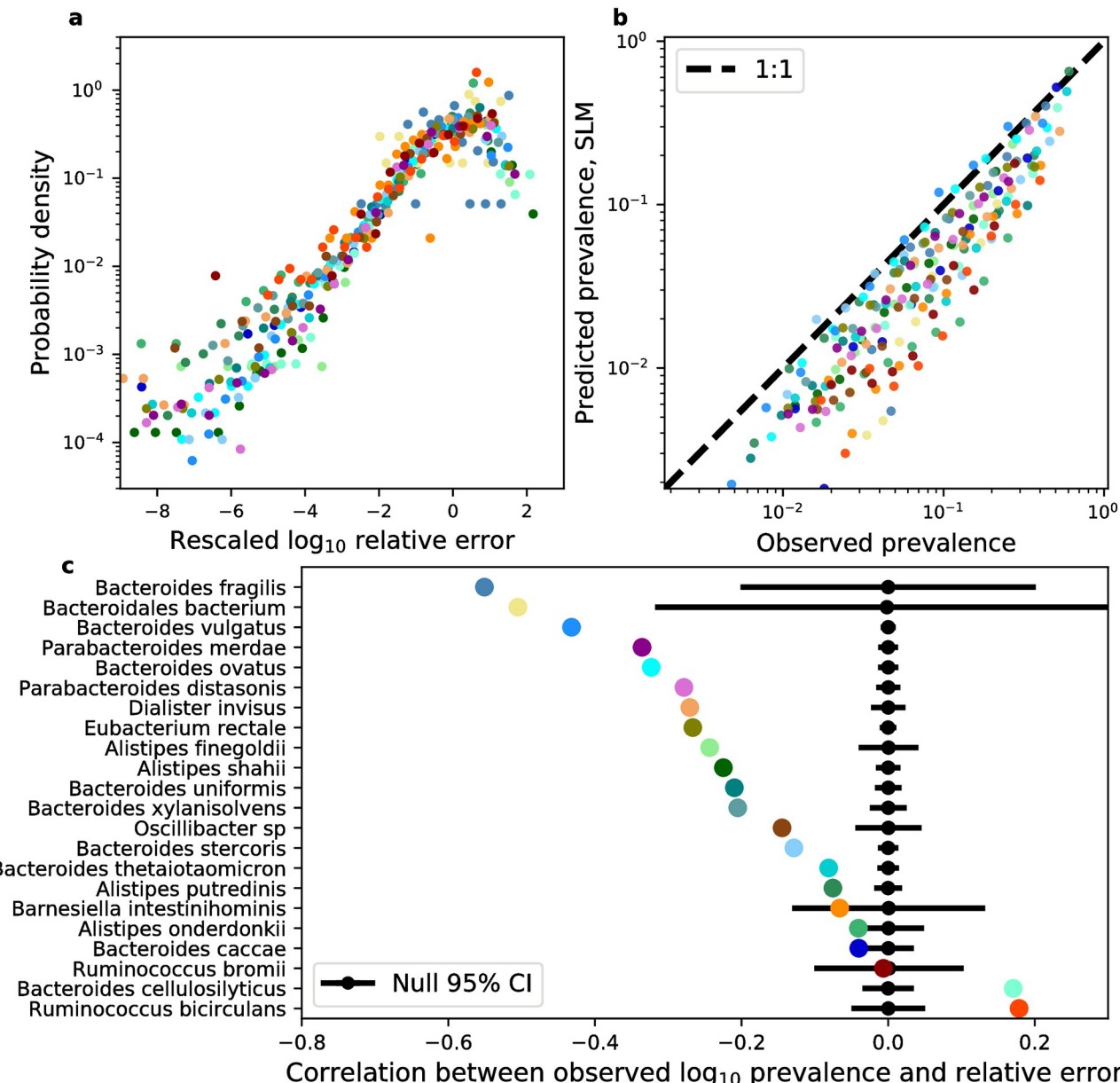

**Fig 3. The SLM succeeds and fails in a consistent manner across phylogenetically distant species. a)** Distributions of logarithmic relative errors of prevalence obtained from Eqs 6 and 21 were rescaled by their mean and standard deviation to illustrate their similarity across species. **b)** Binning observed prevalences reveals how the predicted values tend to follow a one-to-one relationship (dashed black line) across species, with variation among species. **c)** By calculating the correlation between $\log_{10}$-transformed observed prevalence and the relative error of our predictions, one finds a negative correlation for the majority of species. A permutation test confirms that these negative correlations are significant (95% CIs as black lines) for the majority of species (16/22). All analyses in this plot were performed using synonymous sites. Identical analyses and equivalent results for nonsynonymous sites can be found in S11 Fig.

compete, resulting in alternative stable states. This connection allows one to rule out competitive exclusion as probably contributor towards the predictive error.

Alternatively, the comparatively poor performance of an ecological model among low prevalence alleles provides room for evolutionary explanations. Low prevalence alleles may be present due to the evolutionary dynamics operating within a host, where a given allele

can arise in a host due to mutation, increasing the observed prevalence to a value higher than that predicted by an ecological model of strain dynamics. Without a model describing the dynamics of both the ecological and evolutionary dynamics of $f$ within a host it is difficult to parse alleles into discrete "ecological" and "evolutionary" categories. Regardless, the distributions of prevalence prediction errors have a strikingly bimodal form for several species (S7 and S8 Figs), suggesting that there may be some truth to the claim that evolution, rather than ecology, can disproportionately contribute to the prevalence of alleles across hosts.

## Strain-level ecology determines the accuracy of the gamma distribution

While the gamma distribution succeeds at explaining patterns of genetic diversity for an appreciable number of sites across microbial species, the ability to connect quantitative predictions to empirical distributions ultimately rests on the fact that the SLM predicts a distribution that is fully parameterized by the mean and variance of $f$ (Eq 3). Alternatively stated, one does not directly estimate the carrying capacity, they estimate the mean frequency, the expectation of which is a function of the carrying capacity under the SLM that reduces to a proportionality when Taylor's law holds [44]. This reliance on statistical moments estimated from observational data suggests that alternative Langevin equations that can also predict a gamma distribution are equivalent candidates to the SLM in the absence of additional evidence.

In contrast, models of ecology and evolution that predict probability distributions other than the gamma are inappropriate on the outset, as the form of the gamma distribution that explicitly considers the effect of sampling (Eq 4) succeeded in predicting the prevalence of alleles of moderate-to-high mean frequency across hosts (Eq 6). For example, models of neutral evolution $\frac{\partial f}{\partial t} \propto \sqrt{\frac{f(1-f)}{N}} \cdot \eta(t)$ and neutral ecology $\frac{\partial f}{\partial t} \propto \sqrt{f} \cdot \eta(t)$ predict that Fig 1c should resemble a Gaussian distributions over short timescales, a prediction that is incompatible with the observed distribution of within-host allele frequencies across hosts [76, 77] (Fig 1c). In addition, models that predict a lognormal distribution of within-host allele frequencies across hosts such as an ecological model of a strain with a constant rate of growth (as opposed to the logistic growth term in the SLM) and environmental noise (the same noise term as that in the SLM) are also inappropriate since a lognormal distribution did a poor job explaining the empirical distribution [78] (Fig 1c).

Given that the set of potential Langevin equations is constrained to those capable of returning a gamma distribution, I evaluated two evolutionary models that predict a gamma distribution to determine whether they were viable alternatives to the SLM. Starting with a Langevin equation, the frequency dynamics of a single allele are governed by forward and backward mutation ($\mu$, $v$), selection ($s$), and random genetic drift for a population of size $N$.

$$\frac{\partial f}{\partial t} = sf(1-f) + \mu(1-f) + vf + \sqrt{\frac{f(1-f)}{N}} \cdot \eta(t) \tag{7}$$

To reduce the number of free parameters and remove nonlinear terms, one can examine Eq 7 in the low frequency limit ($f \ll 1$) and obtain Langevin equations for evolution under

positive ($s > 0$) and purifying selection ($s < 0$)

$$\frac{\partial f}{\partial t} = -sf + \mu + \sqrt{\frac{f}{N}} \cdot \eta(t) \tag{8a}$$

$$\frac{\partial f}{\partial t} = sf + \mu + \sqrt{\frac{f}{N}} \cdot \eta(t) \tag{8b}$$

A gamma-distributed stationary solution can be derived in the case of purifying selection ($s < 0$; [76, 79]), whereas the stationary solution of Eq 8a is straightforward to derive. For the $s > 0$ case, a gamma distribution of allele frequencies can be derived where the time-dependence is captured by the maximum frequency that an allele can reach ($f_{max} \equiv \frac{e^{st}-1}{2Ns}$) which, in principle, can be estimated from empirical data (Materials and methods). This dynamic form of mutation-selection balance is a gamma distribution that is solely parameterized by the maximum obtainable frequency of said allele ($f_{max}$) and the population scaled mutation rate ($2N\mu$) [25]. Together, these selection regimes provide two forms of mutation-selection balance that predict the gamma distribution (S1 Text).

$$f_{\text{Evo},s<0} \sim \text{Gamma}(2N\mu, 2N|s|) \tag{9a}$$

$$f_{\text{Evo},s>0} \sim \text{Gamma}(2N\mu, f_{max}^{-1}) \tag{9b}$$

Both of these distributions can be parameterized using the mean and variance, where $\langle f \rangle = \frac{\mu}{|s|}$ and $\beta = \frac{1}{2N\mu}$ for $s < 0$ and $\langle f \rangle = 2N\mu f_{max}$ and $\beta = (2N\mu)^2$ for $s > 0$. These distributions provide alternative explanations for the predictive capacity of the gamma distribution, and it is worth investigating their feasibility.

There is evidence that purifying selection is widespread in the human gut microbiome [3, 39], though it is unlikely responsible for the range of across-host allele frequency fluctuations that could be inferred in this study. After accounting for sequencing error and total depth of coverage using MAPGD, the mean of the lowest inferred non-zero allele frequency across all species was $\sim 0.02$. Given that microbial populations in the human gut are typically very large in size, it is unlikely that the same allele independently reached frequencies $\gtrsim 0.02$ in multiple hosts under negative selection. Assuming that independence among sites holds, the frequencies of individual alleles should not exceed $\frac{1}{N|s|}$ [80], meaning that it is unlikely that a substantial fraction of the alleles with non-zero inferred frequencies were driven by purifying selection. This explanation becomes even less likely when one considers the negative relationship between observed allelic prevalence and the accuracy of the gamma.

The dynamic mutation-selection balance parameterization of the gamma distribution also contains forward mutation, a useful feature given that mutation can contribute towards the total observed frequency of a beneficial allele that is increasing in frequency within a given host (Eqs 9a and 9b). Such a feature is appealing, as, when the SLM fails, it does so by under-predicting observed prevalence. However, while the inclusion of forward mutation may increase the predicted value of prevalence, it is unlikely that positive selection substantially contributes towards the typical across-host single-site frequency spectra for alleles of moderate-to-high mean frequency. Furthermore, the explicit time-dependence in the parameterization of $f_{max}$ implies that a given allele is increasing in frequency in the $f \ll 1$ regime in all hosts where non-zero frequencies were observed. This model is highly useful for describing the dynamics of an ensemble of populations where the initial frequency is known and of low frequency, but it is likely inapplicable to single-timepoint samples across unrelated human hosts.

To the extent that this parameterization holds, it likely does so for alleles that are observed at low frequencies, restricting the model's applicability to low prevalence alleles. Because the SLM tends to fail for this prevalence regime, it is possible that a distribution derived from a model of evolutionary dynamics, gamma or otherwise, is necessary to predict genetic diversity among low prevalence alleles.

Beyond the consideration of evolutionary models, it is ultimately necessary to evaluate the extent that the success of the gamma is due to the existence of strain structure. This is a difficult task given that there is no guarantee that a given allele is present in multiple hosts because it is on the background of genetically identical strains. This limitation arises due to the difficulty inherent in determining whether a given host harbors a given strain from a single static metagenomic sample. Such difficulty persists in part due to technical and statistical limitations, but also due to the lack of practical strain definitions [81–84]. It is difficult to phase strains from short-read sequencing data where physical linkage between variants cannot be established, making it necessary to instead identify a single haplotype within a given host [3, 11]. So, while it is currently possible to identify the prevalence of dominant lineages across hosts, there is no straightforward approach that allows one to assign an observed allele to a given strain.

Given these methodological constraints, I identified the existence of strain-level structure for a given species within a given host using `StrainFinder`, a program that infers the number of strains using the shape of the site-frequency spectrum (Materials and methods; [85]). I found that the fraction of hosts harboring strain structure ranged from 0.089–0.64 among the species I examined, with a median of $\sim 0.27$ (Fig 4a). If alleles with higher prevalence were driven by the presence of strain structure, then one would expect a positive correlation between the accuracy of the predicted allele prevalence and the fraction of hosts containing strain structure across species, equivalent to observing a negative correlation between relative error and the fraction of hosts containing strain structure. This prediction held, as the correlation between these two variables was typically negative within a given range of prevalence values, but tended to become more negative when only alleles with high prevalence estimates were included (Fig 4b). To resolve this relationship, I examined the degree of correlation between the relative error of the gamma and the fraction of hosts with strain structure across a

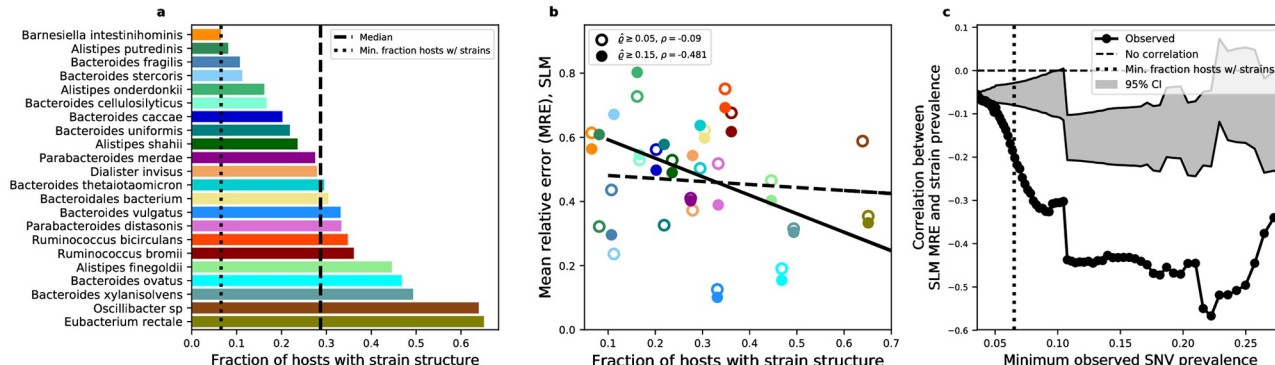

**Fig 4. The presence of strain structure is correlated with the accuracy of the SLM. a)** The presence or absence of strain structure was inferred from the distribution of allele frequencies for each species within each host using `StrainFinder`, providing an estimate of the fraction of hosts that harbor strain structure. **b)** This per-species estimate of strain structure can be compared to the mean relative error of allelic prevalence predictions obtained using the SLM (Eq 6) to determine whether the success of the SLM is correlated with the existence of strains. By examining this relationship when sites with rare alleles (i.e., low prevalence) are included ($\hat{\varrho} \geq 0.05$: dashed line) and excluded ($\hat{\varrho} \geq 0.15$: solid line), one sees a stronger correlation ffor alleles with a high prevalence threshold. **c)** This trend can be systematically evaluated by calculating the correlation across a range of prevalence values (black dots). A permutation test establishes 95% confidence intervals of the null (grey window).

wide range of observed prevalence thresholds (Fig 4c). The correlation tended to increase with prevalence, though there was a substantial decrease once a prevalence threshold of $\sim 0.1$ was reached. The fraction of hosts with strain structure for a given species provide context as they place lower bounds on the range of allelic prevalences that can be driven by strain-level ecology, meaning that alleles with prevalence values lower than the lowest observed fraction of hosts with strain structure are unlikely to be driven by ecology. Given that the species with the lowest observed fraction of hosts with strain structure were close to this descent, it is possible that this value is where the ecological dynamics of strains began to predominantly influence across-host patterns of genetic diversity.

## Discussion

This study demonstrated that a model of ecology that explained the dynamics of strains within a single host can be successfully applied to explain across-host patterns of diversity at individual nucleotide sites, the constituent of strains. I identified patterns of genetic diversity in the human gut microbiome that were statistically invariant across evolutionarily distant species. Motivated by these results, and the prominence of strain structure in the human gut, I identified a prospective model of ecological dynamics (i.e., the Stochastic Logistic Model [44]) that could explain said patterns. Using this model, I was able to predict the fraction of hosts that harbored a given allele (i.e., prevalence) using measurable parameters (i.e., zero statistical fitting) for a considerable fraction of sites across species. Prediction accuracy tended to improve among more prevalent alleles, a result that is consistent with the conceptual picture that both ecology and evolution are operating within a given species in the human gut [10]. The accuracy of prevalence predictions were correlated with independent estimates of strain structure, providing additional empirical evidence that patterns of genetic diversity across human hosts are driven by strain-structure for a considerable fraction of sites.

The success of the SLM for common alleles implies that one's level of sampling (i.e., sequencing coverage) is the primary determinant of whether or not strain structure can be detected within a healthy human host for several species [44]. A lack of genuine absences of strain structure (i.e., extinction) in healthy human hosts would subsequently imply that competitive exclusion is rare at the strain level. This is a strong claim and proving it is beyond the scope of this study. Instead, it is worth noting that the claim seemingly contrasts with the observation that strains exhibit varying frequencies across hosts for several species [3], though fluctuations across hosts alone are not demonstrative of competitive exclusion. Like any model with stochasticity, fluctuations across hosts are expected under the SLM and some number of absences will inevitably arise due to the finite nature of sampling. However, it is also possible that the carrying capacity of a given strain could vary from host-to-host for several species, widening the across-host distribution of frequencies (Fig 1c). This detail can readily be incorporated into the SLM if one assumes that the carrying capacity in a given host is an independently drawn random variable from some unknown distribution [67]. Thus, given certain assumptions, the success of the SLM is reconcilable with the view that the carrying capacity of a strain is host-dependent.

While the SLM successfully predicted the prevalence of common alleles across hosts and has been shown to describe the temporal dynamics of strains within a host [46], all models have limitations, and it is useful to briefly discuss those that are applicable. A fundamental limitation of the SLM is that it is phenomenological in nature. It is unclear how microscopic details that are relevant to the ecological dynamics of strains, namely, consumer-resource dynamics [20, 40, 86, 87], map onto the SLM. Models that incorporate such dynamics are capable of recapitulating the temporal dynamics of microbial communities at the species level

[88], and are necessary to model the emergence of new strains and their subsequent eco-evolutionary dynamics [87], suggesting that these microscopic details may be necessary to describe certain macroecological patterns at the strain level. As a contrast, the phenomenological nature of the SLM could be viewed as an asset when one wants to capture and predict multiple empirical patterns using an analytic solution without the use of fitted parameters. Indeed, the SLM succinctly captures the dynamics of a constrained random walk [88] and it is likely that alternative models of strain-level ecology that capture the same stochastic process are equally applicable.

The conclusion that properties of a substantial fraction of alleles can be predicted across hosts using a model of ecology, as opposed to evolutionary, dynamics is of consequence to studies of diversity in the human gut. Measures of genetic diversity within a single host (e.g., nucleotide diversity) are often used to assess the genetic content of a microbial species [12, 89–92]. Recent efforts to characterize patterns of genetic diversity within a single host have found that the temporal dynamics of nucleotide diversity are primarily driven by fluctuations in strain frequencies over time [46]. In addition, the contribution of strain structure to estimates of genetic diversity from unrelated human hosts has been previously reported [3]. This study builds on past results by specifying that strain structure shapes patterns of genetic diversity across hosts. The implications of strain-level ecology likely extends to measures of genetic differentiation between populations (e.g., fixation index, $F_{ST}$) that have been used to assess the degree of structure of a given species across human hosts [12]. The results presented in this study suggest that single-sample measures of genetic diversity that do not account for strain ecology are unlikely to be informative of evolutionary processes operating within the human gut.

The success of a prediction is often contingent on one's range of observation. After accounting for sequencing error, the lowest inferred allele frequencies ranged from 0.006–0.06 across species with an average of $\sim 0.02$. A straightforward calculation suggests that this range is higher than the true minimum frequency (i.e., $1/N$) by several orders of magnitude. The mean relative abundances of a given species across hosts ranged from 0.02–0.1. Previously established order-of-magnitude estimates of the typical number of cells in the human gut range from $\sim 10^{13} – 10^{14}$, from which one can use the mean relative abundance distribution across species to calculate a first-pass range of empirical abundances of $\sim 10^{11} – 10^{13}$ [1, 93]. This range suggests that the true minimum frequency of an allele is at least eight orders of magnitude lower than the minimum inferred allele frequency. It is clear from this calculation that, at present, researchers are only able to examine a narrow range of possible allele frequencies within the human gut microbiome, a range that is likely primarily driven by strain structure.

The implication of this relatively narrow observational window is that the success of predictions derived from ecological principles, and the feasibility of alternative single-locus models of evolution, are likely contingent on present limitations on the depth and error rates of shotgun metagenomic sequencing. As advances in sequencing technology and statistical inference continue to permit lower observational thresholds and provide information about physical linkage between variants [5, 94–97], one expects that an increasingly higher fraction of observed alleles will be subject to evolutionary processes rather than the ecological processes affecting the strain harboring said allele, reducing the aptness of ecological models. Succinctly stated, the existence of strain structure suggests that the dynamics of allele frequencies are likely dependent on the frequency scale at which observations can be made. By expanding said range, it may be possible to identify a frequency threshold where observable alleles are primarily driven by evolutionary dynamics rather than the ecological dynamics of strains, providing the means to test quantitative predictions of recent developments in population genetic theory

[25, 80]. Recognition of the possibility of such scale-dependence has the potential to shape future studies and rigorously assess the purported universality of empirical patterns of genetic diversity in the human gut microbiome.

Throughout this manuscript I have assumed that strains are sufficiently genetically diverged such that within-host structure is overt (i.e., many alleles with intermediate frequencies within a host, $0 < f < 1$). However, *de novo* strains can emerge within a host, resulting in within-host structure where strains are separated by only a handful of SNVs (e.g., *Bacteroides fragilis* strains in [1]). It is worth considering how the emergence of new strains relates to the patterns documented in this manuscript. A recently diverged strain within a single host is analogous to a species that is only found in a single host. Given that the sampling form of the gamma distribution used in this study succeeded at predicting the prevalence of species present in a single host [44], it should, in principle, also succeed in predicting the prevalence of a SNV observed in a single host due to recently emerged strain structure. This expectation is not the case, as predictions for low prevalence alleles consistently failed for all species included in this study (Figs 2 and 3, S5, S6 and S11 Figs). It is reasonable to interpret this lack of predictive success for low prevalence alleles as a consequence of said alleles being present in a low number of hosts due to evolutionary dynamics, rather than their presence being a reflection of newly emerged within-host strain structure. Though this interpretation does not mean that recently diverged strains are absent in this cohort of human hosts. Rather, it is instead likely that the macroecological lens applied in this study has insufficient resolution to identify alleles that reflect recently diverged strains that have colonized a low number of hosts, a number similar to the number of hosts in which we would expect to observe a given allele due to evolutionary dynamics (e.g., recurrent mutation).

Finally, it worth commenting on the applicability of these results towards evaluating the ecological effects of environmental perturbations, namely, host-induced changes. Across-host patterns are often the consequence of within-host dynamics, so, in principle, it should be possible to use the SLM to predict across-host changes in response to perturbations. However, it is difficult to know *a priori* whether a host is in a perturbed state unless the perturbation is administrated as part of a controlled study (e.g., major diet change, drug trials, etc.). For example, one could compile data from studies on different human hosts where courses of antibiotics were administered and metagenomic sequencing was performed over time. Knowing the inferred frequency of a strain or the frequencies of its constituent alleles at the start of the trial $f_0$, one could then leverage the SLM and its stationary solution to determine when statistical quantities calculated across hosts such as the mean frequency or prevalence reach their stationary values as the system relaxes away from its perturbed state (i.e., $\langle f | t, f_0 \rangle \rightarrow \langle f \rangle$ or $\langle \varrho | t, \varrho_0 \rangle \rightarrow \langle \varrho \rangle$).

## Materials and methods

### Data acquisition and processing

To investigate patterns of genetic diversity within the human gut microbiome, I used shotgun metagenomic data from 468 healthy North American individuals sequenced by the Human Microbiome Project [24, 98]. I first processed the data using a previously developed analysis pipeline to identify the set of sites in core genes [3]. This pipeline uses a standard reference-based approach (MIDAS v1.2.2 [29]) to map reads from each metagenomic sample to reference genes across a panel of prevalent species and filter reads based on quality scores and read mapping criteria. Definitions of "species" vary across disciplines in biology. To avoid ambiguity, I opted for a direct operational definition provided by the resolution of the reference genome

panel used by `MIDAS`, a definition that has been used in many studies of the human gut microbiome [3, 5, 25, 36, 39, 46, 99].

The relative abundance of each species in a given host was inferred using `merge_midas.py species`. Then, the command `merge_midas.py genes` was run with the following flags: `--sample_depth 10`, `--min_samples 1`, and `--max_species 150`. Using this processed gene output, the command `merge_midas.py snps` was run with the following flags: `--sample_depth 5`, `--site_depth 3`, `--min_samples 1`, `--max_species 150`, and `--site_prev 0.0`. I only processed samples from unrelated hosts and removed all temporal samples, leaving us with a set of samples that correspond to observations from unique hosts. I retained alleles in sites that were present in a core gene (i.e., a gene that was present in $\geq$ 90% of hosts) with a minimum total within-host depth of coverage of 20 ($D \geq 20$) in at least 20 hosts. These parameter settings are effectively identical to the settings used in prior studies that have examined the HMP [3, 7, 39, 46]. The (non)synonymous status of sites were determined using `MIDAS` reference genomes. While members of a species can differ in genomic content and can alter the accuracy of calling the status of a site, this pipeline was previously run on the same dataset used in this manuscript to test population genetic predictions on measures of genetic diversity using nonsynonymous and synonymous sites, finding that the empirical data matched theoretical predictions [3], implying that any ambiguity in site (non)synonymous assignment did not shape our results.

After identifying the set of sites in core genes that passed the quality control thresholds, I obtained appropriate BAM files so that allele frequencies could be reliably estimated. First, using the reference genomes used by `MIDAS` and the BAM file generated by `MIDAS` for all species in a single host, I split the BAM file containing all species into separate BAM files for each species using the command `samtools view` with default settings [100]. Each BAM file was sorted using the `samtools sort` command with default settings, from which a `.header` file was using `samtools view`.

Sequencing errors can obfuscate naïve estimates of low frequency alleles ($f \ll 1$). This effect is particularly on measures of genetic diversity, as sequencing error-induced noise can easily swamp real biological signal when statistics are calculated over a large number of sites. Studies often attempt to control for errors by restricting their analyses to sites that pass a particular threshold for the total depth of coverage and/or the coverage of the minor allele. However, the effectiveness of an arbitrary cutoff will not be identical across sites and hosts due to variance in the total depth of coverage. For this specific analysis, reliable frequency estimates are also necessary since an allele of any non-zero frequency can contribute towards measures of prevalence, so it is imperative for low frequency alleles to be estimated in a statistically justified and unbiased manner that balances sensitivity and specificity.

To account for sequencing errors, I elected to use the full maximum likelihood estimator `MAPGD` v0.4.40 [101], an unbiased estimator that accounts for the total depth of coverage and an unknown sequencing error. This estimator was chosen due its comparative high sensitivity to low frequency alleles without sacrificing its false discovery rate in real and simulated data [102], reflecting a balance between sensitivity and specificity. Using BAM and `.header` files processed from `MIDAS` output, `MAPGD` was run with a log-likelihood ratio polymorphism cutoff of 20 (`-a 20`), a choice informed by prior benchmarking studies and `MAPGD` recommendations [102]. The choice of log-likelihood ratio cutoff is unlikely to shape the results of this study, as the cutoff effectively establishes a coverage cutoff which can be incorporated into predictions derived using probability distributions that explicitly account for sampling (e.g., Eq 4). Samples with insufficient coverage for `MAPGD` to be run were removed from all downstream analyses. I then polarized alleles based on the major allele across hosts.

Gamma and lognormal distributions were fit to the distribution of within-host allele frequencies across all hosts using `SciPy`. Because the distribution was rescaled using the logarithm of the data, the distributions were fit as if I was interested in the logarithm of the random variable (i.e., $\log_{10}(f)$). This detail translates to fitting the loggamma instead of the gamma and the Gaussian instead of the lognormal. AIC was calculated using custom scripts.

The `MAPGD` inference procedure makes no assumptions about the existence of strain structure. If strain structure is present in the data it will shape the distribution of allele frequencies, subsequently altering measures of genetic diversity and the predictive capacity of the SLM. To determine whether the error of the SLM was related to the existence of strains, I used an algorithm to determine whether strain structure was present for each species within each host. As an independent estimate of strain structure, I estimated strain frequencies for all species using all sites with $\geq 20$ fold coverage within each host by applying `StrainFinder` v1.0 [85] to the frequency spectra obtained from the upstream pipeline [3]. The program `StrainFinder` was run on each sample for each species using 10 initial conditions using local convergence criteria with the following flags: `--dtol 1`, `--ntol 2`, `--max_time 20000`, `--converge`. The program was run for strain numbers ranging from one to four the estimates with the top five log-likelihoods were retained. I then selected strain frequencies with the lowest Bayesian Information Criterion for each species in each sample. Joint density plots (e.g., Fig 2) were made using functions from `macroecotools` v0.4.0 [103].

## Predicting allele prevalence using the SLM

**Deriving the distribution of allele frequencies from the SLM.** We begin with the assumption that the typical polymorphism observed within a given host for a given species is present because it is on the background of a colonizing strain. In such a scenario, the dynamics of an allele are not determined by its evolutionary attributes (i.e., fitness effect, mutation rate, etc.) but by the ecological dynamics of the strain. There is increasing evidence that the stochastic logistic model of growth (SLM) is a suitable null model of microbial ecological dynamics at the species level [44, 66, 67] and recent evidence indicates that the SLM sufficiently fits the temporal dynamics of strains within a human host for the vast majority of microbial species [46]. A non-trivial application of the SLM to strain-level ecology requires there to be more than one strain within a given host, giving the allele a range of frequencies of $0 < f < 1$

$$\frac{\partial f}{\partial t} = \frac{f}{\tau_i}\left(1 - \frac{f}{K_i}\right) + \sqrt{\frac{\sigma_{\tau_i}}{\tau_i}}f \cdot \eta(t) \tag{10}$$

The terms $\frac{1}{\tau_i}$, $K_i$, and $\sigma_{\tau_i}$ represent the intrinsic growth rate of the strain, carrying capacity, and the coefficient of variation of growth rate fluctuations. The term $\eta(t)$ is a Brownian noise term where $\langle\eta(t)\rangle = 0$ and $\langle\eta(t)\eta(t')\rangle = \delta(t - t')$ [69]. By definition, strain frequencies within a species must be between zero and one, so $0 < K_i < 1$.

Using the Itô $\leftrightarrow$ Fokker-Planck equivalence [69], one can formulate a partial differential equation for the probability $p(f, t)$ that an allele has frequency $f$ at time $t$

$$\frac{\partial p(f, t)}{\partial t} = -\frac{\partial}{\partial f}\left[\left(\frac{f}{\tau_i}\left(1 - \frac{f}{K_i}\right)\right)p(f, t)\right] + \frac{\sigma_{\tau_i}}{2\tau_i}\frac{\partial^2}{\partial f^2}(f^2 p(f, t)) \tag{11}$$

From which one sets $\frac{\partial p}{\partial t} = 0$ to obtain the stationary probability distribution of allele frequencies

$$p(f) = \frac{1}{\Gamma(2\sigma_{\tau_i}^{-1} - 1)} \left(\frac{2}{K_i \sigma_{\tau_i}}\right)^{2\sigma_{\tau_i}^{-1} - 1} \exp\left[-\frac{2}{K_i \sigma_{\tau_i}} f\right] f^{2\sigma_{\tau_i}^{-1} - 2} \tag{12}$$

This distribution, known as the abundance fluctuation distribution in macroecology [44], is a gamma distribution with the following mean and squared coefficient of variation

$$\langle f \rangle = K_i \left(1 - \frac{\sigma_{\tau_i}}{2}\right) \tag{13a}$$

$$\frac{\langle f^2 \rangle - \langle f \rangle^2}{\langle f \rangle^2} = \frac{\sigma_{\tau_i}}{2 - \sigma_{\tau_i}} \tag{13b}$$

Defining empirical estimates of $\langle f \rangle$ and $\frac{\langle f^2 \rangle - \langle f \rangle^2}{\langle f \rangle^2}$ as $\overline{f}$ and $\beta^{-1}$, I obtained a form of the gamma (represented in its shape/rate parameterization form) that can be used to generate ecological predictions of measures of genetic diversity with zero free parameters

$$p(f|\beta, \beta/\overline{f}) = \frac{1}{\Gamma(\beta)} \left(\frac{\beta}{\overline{f}}\right)^{\beta} \exp\left[-f\frac{\beta}{\overline{f}}\right] f^{\beta - 1} \tag{14}$$

**Deriving prevalence predictions using the SLM.** The probability of detecting an allele of a given frequency within a host depends on one's sampling effort. The impact of finite sampling from gamma distributed random variables has been previously examined within macroecology [44], results that I apply and extend in this section. To model this process, I start by letting $(A, D)$ denote the number of reads of the alternate allele and total sequencing depth at a given site. I estimate the frequency of the alternate allele within a given host as $\hat{f} = A/D$. I assume that sampling distribution of $A$ is binomial

$$\Pr[A|D, f] = \binom{D}{A} f^A (1 - f)^{D - A} \tag{15}$$

When $D \gg 1$ and $f \ll 1$ while $D \cdot f$ remains finite, the binomial sampling process can be approximated by the Poisson distribution

$$\Pr[A|D, f] = \frac{(D \cdot f)^A}{A!} e^{-D \cdot f} \tag{16}$$

Using this approximation, one can solve the integral for the probability of observing $A$ reads assigned to the alternate allele out of $D$ total reads when $f$ is a gamma distributed random

variable

$$\Pr[A|D, \beta, \beta/\overline{f}] \quad = \quad \int_0^1 \Pr[A|D,f] \cdot p(f|\beta, \beta/\overline{f})\, df \tag{17a}$$

$$= \quad \frac{D^A}{A!\Gamma(\beta)\frac{\beta^\beta}{\overline{f}}} \int_0^1 f^{A+\beta-1} e^{-f\left(D+\frac{\beta^{-1}}{\overline{f}}\right)}\, df \tag{17b}$$

$$= \quad \frac{\Gamma(\beta+A)}{A!\Gamma(\beta)} \left(\frac{\overline{f}D}{\beta+\overline{f}D}\right)^A \left(\frac{\beta}{\beta+\overline{f}D}\right)^\beta \tag{17c}$$

The distribution $\Pr[A|D, \beta, \beta/\overline{f}]$ now represents a gamma distribution that explicitly accounts for sampling. By setting $A = 0$, one can calculate the probability of not detecting the alternate allele (i.e., absence) with a sampling depth of $D$ reads [44, 71]

$$\Pr[0|D, \beta, \beta/\overline{f}] = \left(1 + D \cdot \frac{\beta}{\overline{f}}\right)^{-\beta} \tag{18}$$

From which one can calculate the expected prevalence of the allele $\langle \varrho \rangle$ over $M$ hosts as

$$\langle \varrho \rangle = \frac{1}{M} \sum_{m=1}^{M} (1 - \Pr[0|D_m, \beta, \beta/\overline{f}]) \tag{19}$$

## Evaluating prevalence predictions

Full derivations of the predicted prevalence of each model can be found in the Materials and Methods. The predicted values of prevalence were compared to the following estimate of observed prevalence.

$$\hat{\varrho} = \frac{1}{M} \sum_{m=1}^{M} (1 - \delta_{f_m,0}) = 1 - \frac{1}{M} \sum_{m=1}^{M} \delta_{f_m,0} \tag{20}$$

where $f_m$ is the frequency of the alternative allele in the $m$th host and the Kronecker delta $\delta_{f_m,0}$ is equal to 1 if $f_m = 0$ and zero otherwise. To evaluate the success of the predictions I calculated the relative error of a given prediction

$$\varepsilon = \left| \frac{\text{Obs.} - \text{Pred.}}{\text{Obs.}} \right| \tag{21}$$

I performed permutation tests to determine whether the SLM had higher success among alleles with higher prevalence. By permuting all values of $\varepsilon$ for a given species and calculating the Pearson correlation coefficient between it and the observed prevalence, I obtained a null distribution of correlation coefficients from which I calculated 95% intervals.

To determine whether there was a relationship between the error of the SLM and the fraction of hosts with strain structure across species, I implemented a permutation approach. First, for each species, I calculated the number of alleles in a given prevalence threshold ($T$ total thresholds). I then permuted all values of $\varepsilon$ and calculated the mean $\varepsilon$ using the number of

alleles that were found in each prevalence threshold, $(\overline{\varepsilon}_1, \cdots, \overline{\varepsilon}_T)$. The correlation coefficient was then calculated between $\overline{\varepsilon}_t$ and the fraction of hosts containing strains among all species for each prevalence threshold, allowing us to obtain a null distribution of correlation coefficients for all values of $t$. I only retained a prevalence threshold for a given species if there were at least 10 sites within the threshold.

## Supporting information

**S1 Fig. Measures of genetic diversity among nonsynonymous sites.** Measures of genetic diversity calculated from nonsynonymous sites exhibit similar statistical forms across phylogenetically distant species in the human gut, similar to patterns observed among synonymous sites (Fig 1).
(TIF)

**S2 Fig. AFD survival curves for synonymous sites.** Survival forms of rescaled distributions of within-host allele frequencies across hosts and mean frequencies across hosts. Representing the data presented in Fig 1c and 1d reveals how distributions of genetic diversity have similar forms across phylogenetically distant species. Each non-black line represents a species. A dashed black line represents the fit of a gamma distribution and dotted black line represents a lognormal.
(TIF)

**S3 Fig. AFD survival curves for nonsynonymous sites.** The equivalent plot for S2 Fig for nonsynonymous sites.
(TIF)

**S4 Fig. Coverage distribution.** The use of the log-likelihood ratio in MAPGD introduces a lower bound on the total depth of coverage ($D$) necessary to estimate the frequency of an allele at a given site. **a)** The existence of a lower bound translates to a truncation of the data, where I did not observe any sites with a coverage less than 20 that were processed by MAPGD. **b,c)** This truncation means that the depth of coverage of a minor allele ($A$) cannot be less than half the total coverage (e.g., 10).
(TIF)

**S5 Fig. Synonymous prevalence predictions.** A direct comparison between the observed prevalence of all alleles and their corresponding predicted prevalences using the SLM for synonymous sites. A total of 1,000 datapoints were sampled without replacement for each subplot.
(TIF)

**S6 Fig. Nonsynonymous prevalence predictions.** Analogous analyses to S5 Fig using nonsynonymous sites.
(TIF)

**S7 Fig. Synonymous prevalence prediction error distributions.** By calculating the relative error of all alleles for the SLM I can examine the error distributions across species. To visually compare the two models, I examined the survival distribution of the relative errors (i.e., the compliment of the empirical cumulative density function). All alleles in this plot are at synonymous sites.
(TIF)

**S8 Fig. Nonsynonymous prevalence prediction error distributions.** Analogous analyses to S7 Fig using nonsynonymous sites.
(TIF)

**S9 Fig. Synonymous relationship between $\bar{f}$ and prevalence.** The empirical relationship between the mean frequency of an allele ($\bar{f}$) and its prevalence across hosts can be recapitulated by the SLM for synonymous sites. Blue dots represent observed values and the shade of blue is proportional to the density of observations. The black line is the predicted relationship calculated using Eq 11. A total of 1,000 datapoints were sampled without replacement for each subplot.
(TIF)

**S10 Fig. Nonsynonymous relationship between $\bar{f}$ and prevalence.** Analogous analyses to S9 Fig using nonsynonymous sites.
(TIF)

**S11 Fig. Nonsynonymous prevalence error analysis.** The equivalent analyses in Fig 3 were performed on alleles at nonsynonymous sites. The results of these analyses are qualitatively consistent with those of synonymous sites.
(TIF)

**S12 Fig. Relationship between $\bar{f}$ and $\beta$ for synonymous sites.** The relationship between the empirical estimates of the two parameters of the SLM: the mean allele frequency across hosts ($\bar{f}$) and the squared inverse of the coefficient of variation of frequencies across hosts ($\beta$). Each point is an individual allele. All alleles are on synonymous sites. A total of 1,000 datapoints were sampled without replacement for each subplot.
(TIF)

**S13 Fig. Relationship between $\bar{f}$ and $\beta$ for nonsynonymous sites.** Analogous analyses to S12 Fig using nonsynonymous sites.
(TIF)

**S1 Text. Supplemental information.** Derivation of the distribution of allele frequencies under a linearized single-locus model of evolution.
(PDF)

## Acknowledgments

I thank S. Bald and R.W. Wolff for their assistance with `StrainFinder` and both N.R. Garud and R.W. Wolff for their comments on an early draft. Thanks to B.H. Good for pivotal discussions, sharing their insights, and for making their lecture notes available to the public. Thanks to S. Bubnovich, J. Grilli, D. Reyes-González, and N.I. Wisnoski for their feedback on the manuscript. Finally, thanks to M.S. Ackerman for their assistance with `MAPGD`. This work used computational and storage services associated with the Hoffman2 Shared Cluster provided by UCLA Institute for Digital Research and Education's Research Technology Group.

## Author Contributions

**Conceptualization:** William R. Shoemaker.

**Data curation:** William R. Shoemaker.

**Formal analysis:** William R. Shoemaker.

**Funding acquisition:** William R. Shoemaker.

**Investigation:** William R. Shoemaker.

**Methodology:** William R. Shoemaker.

**Project administration:** William R. Shoemaker.

**Resources:** William R. Shoemaker.

**Software:** William R. Shoemaker.

**Supervision:** William R. Shoemaker.

**Validation:** William R. Shoemaker.

**Visualization:** William R. Shoemaker.

**Writing – original draft:** William R. Shoemaker.

**Writing – review & editing:** William R. Shoemaker.

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
