## [Decision Letter · Decision Letter 0]

6 Apr 2023

PONE-D-22-30239A macroecological perspective on genetic diversity in the human gut microbiomePLOS ONE

Dear Dr. Shoemaker,

Thank you for submitting your manuscript to PLOS ONE. After careful consideration, we feel that it has merit but does not fully meet PLOS ONE’s publication criteria as it currently stands. Therefore, we invite you to submit a revised version of the manuscript that addresses the points raised during the review process. In particular, there are a number of writing issues to be addressed, and how the manuscript presents new results that contribute to the field.

We look forward to receiving your revised manuscript.

Kind regards,

Karthik Raman, Ph.D.

Academic Editor

PLOS ONE

“This work was supported by the NSF Postdoctoral Research Fellowships in Biology Program under Grant No. 2010885 (W.R.S.). This work used computational and storage services associated with the Hoffman2 Shared Cluster provided by UCLA Institute for Digital Research and Education’s Research Technology Group.”

“This work was supported by the NSF Postdoctoral Research Fellowships in Biology Program under Grant No. 2010885 (W.R.S.).

https://beta.nsf.gov/funding/opportunities/postdoctoral-research-fellowships-biology-prfb

3. Please upload a copy of Figure 5, to which you refer in your text on page 17. If the figure is no longer to be included as part of the submission please remove all reference to it within the text.

Additional Editor Comments:

The reviews for the manuscript are in. While the reviewers found interesting aspects of your work, they have raised major concerns. Please revise the manuscript taking into consideration the reviews of both the reviewers.

Reviewers' comments:

Reviewer's Responses to Questions

**Comments to the Author**

1. Is the manuscript technically sound, and do the data support the conclusions?

Reviewer #1: No

Reviewer #2: Yes

2. Has the statistical analysis been performed appropriately and rigorously? 

Reviewer #1: No

Reviewer #2: Yes

3. Have the authors made all data underlying the findings in their manuscript fully available?

Reviewer #1: No

Reviewer #2: Yes

4. Is the manuscript presented in an intelligible fashion and written in standard English?

Reviewer #1: No

Reviewer #2: Yes

5. Review Comments to the Author

Reviewer #1: This manuscript by William Shoemaker explores intraspecies diversity within and across individual microbiomes, with a focus on the applicability of stochastic logistic model and the existence of strain structure. It is my understanding that the main result of this paper is that the result of this modeling is that strains (ecology) are responsible for more within-host variants than mutation (evolution). This result echoes the current thinking in the field, which has been shown before by other papers, including at least one on which this author is a contributor.

This manuscript is difficult to read – partly because what is done is frequently described before what is the goal of this work is, and partly because concepts are a bit muddled / poorly introduced. I list some examples below. I also have some issues with data analysis. Most critically, it is hard to understand what is new in this paper. While it is my understanding that novelty is not a criterion for publication, more fully placing work in the context of the literature is very important for any theoretical or modeling based paper.

Writing issues:

It hard to understand from the abstract, introduction, text, and discussion, exactly is new in this manuscript. The introduction spends much time opining on the value and need for models, without really explaining what has been shown before about the SLM in microbiomes. In particular, the manuscript cites works, including from this author, showing that strains are responsible for most intraspecies dynamics in the microbiome (this has been known for quite some time) and SLMs describe these dynamics well. What was the open question the other set out to solve? If there was some doubt about the applicability of the SLM model and strains prior to this work, what was the critical missing analysis? What required the need for this approach?

It requires a lot of work for a reader to understand what is being referred to at each point in the text. For example:

• At Line 94: “the empirical distribution” is unclear; “the distribution of within-host allele frequencies across all hosts” would be clearer. Even clearer would be “the distribution of within-host frequencies at a given site, across all hosts” or something else.

• Line 339: “unlikely for the allele frequency fluctuations I was able to observe”. Does this refer to fluctuations across hosts, or time? At the very least, a figure panel should be referred to here.

• Line 418: what is a “true strain absence”. Is the author suggesting that single strain colonization is rare for all species and strains? This would conflict with the literature (e.g. B fragilis almost always colonizes humans as a single strain, PMID: 30673701), but if it is supported by the data this should be made more clear.

• Line 104: which variable? Mean and variance across what?

Line 500: I’m not sure exactly what point is trying to be made here. If there is already an interpretation of the data (antibiotics wiped out a strain), how does the SLM help?

Technical notes:

I find Figure 1 (as well as some s to not be presented in a manner such that I can evaluate the claim being made: that the rescaled distributions are similar across species. To evaluate this, I would like to see example distributions (perhaps cumulative curves) drawn for each species – instead what I am presented with is a mess of dots on top of each other in 1b and 1d. I see notable differences, even though these are rescaled already. No goodness of test fits are applied and no alternative distributions are provided.

Site frequency spectra are not widely used in microbiology, as the SFS is something that results from recombination. The idea that within-host or across-host evolution would result in a meaningful SFS is confusing to this reviewer.

Fig 2c: I’m not sure what the value of predicting “mean within-host allele frequency across hosts” is, if all hosts—including those without the allele at all—are included. Perhaps I am missing something. Also, it seems like most of the data here isn’t fitting the line, but instead fitting a curve starting at y = 10^-1. These graphs should be made as density maps to make it clearer how well the data and the prediction line up.

Some thoughts:

It is clear that the author has put a lot of work into the generation of this data and model, but it is always important to be clear about what is known and what is novel. The formalization of an already proposed model can be valuable. Presenting a model which more clearly shows what has been shown by others could also be valuable; if this is the goal than more effort needs to be put into making it clear.

Reviewer #2: In this work, the author used a statistical approach to a publicly available metagenomic dataset to evaluate the dynamics of genetic diversity in the human gut microbiome. They used a proposed distribution of allele frequencies to predict the prevalence of a given allele in a given species in a host. They found that the Stochastic Logistic Model accurately predicted this across species, and could also predict the prevalence of alleles across multiple hosts.

Generally, this is an interesting work. However, as someone with a limited background in biostatistics, I find the description of the underlying modeling approach very hard to follow. For instance, the assumptions behind the SLM described from line 197-208 are not clear to me. Hence, my recommendation is to clarify the description of the SLM, its assumptions, and the overall discussed concepts for readers less familiar with them. I understand that this is not my area of expertise, but it would be very helpful to make the presented results more accessible to scientists in the gut microbiome field.

Minor comments:

- The legends for Figures 1-4 are more a repetition of the Results section than a clear description of what is shown in each panel.

- Line 339: MAGPD is mentioned here for the first time without having been defined before.

- Line 448-449: “Regardless of the speci c model, the conclusion that a substantial fraction of observable alleles are primarily subject to ecological, as opposed to evolutionary, dynamics is of consequence to studies of genetic diversity in the human gut.” It is not clear to me which part of the Results section precisely shows this.

- Line 655: There is a figure legend for Figure 5, which dos not exist.

- Line 704: Is this really the correct URL for the Human Microbiome Project?

6. PLOS authors have the option to publish the peer review history of their article (what does this mean?). If published, this will include your full peer review and any attached files.

Reviewer #1: No

Reviewer #2: No

---

## [Author Response · Author response to Decision Letter 0]

24 Apr 2023

Below is my response that can also be found in the uploaded document 20230420_1538_Response.pdf

Reviewer #1

This manuscript by William Shoemaker explores intraspecies diversity within and across individual microbiomes, with a focus on the applicability of stochastic logistic model and the existence of strain structure. It is my understanding that the main result of this paper is that the result of this modeling is that strains (ecology) are responsible for more within-host variants than mutation (evolution). This result echoes the current thinking in the field, which has been shown before by other papers, including at least one on which this author is a contributor.

Author response: The main result of this paper is that a model of ecology that succeeded at capturing the temporal dynamics of strains within a single host (Wolff et al., 2023), the Stochastic Logistic Model of growth, was capable of quantitatively predicting patterns of allelic prevalence across hosts for a considerable fraction of sites. In the revision I have rewritten portions of the Abstract, Introduction, and Discussion to emphasize these points (Lines 69-72; 73-83; 441-456).

This manuscript is difficult to read – partly because what is done is frequently described before what is the goal of this work is, and partly because concepts are a bit muddled / poorly introduced. I list some examples below. I also have some issues with data analysis. Most critically, it is hard to understand what is new in this paper. While it is my understanding that novelty is not a criterion for publication, more fully placing work in the context of the literature is very important for any theoretical or modeling based paper.

Author response: Interdisciplinary research is often difficult to communicate. In writing the submitted version of this manuscript I meticulously incorporated feedback from peers in macroecology, microbial ecology, statistical physics, and population genetics. I understand that attempting to please too many points of view can lead to a message that is less than precise. I have worked to revise this manuscript to address the Reviewer’s comments. For example, the Abstract now clarifies the purpose of this study and how it fits into the field’s current understanding of strain ecology. The Introduction now explicitly states how we can quantitatively describe the dynamics of strains within a single human host, a result that provides the motivation to use the SLM to investigate patterns of genetic diversity across hosts (Lines 50-83).

By adopting a statistical physics approach and leveraging my past efforts to characterize patterns of diversity within a single host, I was able to 1) successfully predict the prevalence of alleles across hosts for a high proportion of sites and 2) provide evidence that the success of my predictions was due to the existence of strain structure. 

Writing issues:

It hard to understand from the abstract, introduction, text, and discussion, exactly is new in this manuscript. The introduction spends much time opining on the value and need for models, without really explaining what has been shown before about the SLM in microbiomes. In particular, the manuscript cites works, including from this author, showing that strains are responsible for most intraspecies dynamics in the microbiome (this has been known for quite some time) and SLMs describe these dynamics well. What was the open question the other set out to solve? If there was some doubt about the applicability of the SLM model and strains prior to this work, what was the critical missing analysis? What required the need for this approach?

Author response: The question was whether an ecological model that succeeded in predicting macroecological patterns at the species level and in predicting the temporal dynamics of strains within a single host could also predict patterns of genetic diversity across hosts due to the existence of strain structure. 

Making the jump from within-host dynamics to across-hosts patterns is not an easy task at the strain level. Within a given host, the relative abundance of a strain often has to be inferred using the temporal trajectories of individual genetic variants (Roodgar et al., 2021; Wolff et al., 2023). This approach requires several metagenomic sequencing samples within a host over a period time, a fairly high cost that limits the number of hosts that can be sequenced. Therefore, when investigating questions of strain macroecology across a large number of unrelated hosts you often only have access to samples obtained from a single timepoint (e.g., the Human Microbiome Project data examined in this study). In this scenario, it is reasonable to examine individual nucleotide sites and determine whether their patterns follow what you would expect if they were present because they were on a strain, rather than dealing with the difficulty of inferring strain relative abundances from a single metagenomic sample. 

I have revised the Introduction to clarify existing knowledge gaps of strain ecology in the human gut. I now explicitly list patterns that the SLM has previously captured at the species level within and across human hosts (e.g., results described in Grilli, 2020) and how it has succeeded at explaining the temporal dynamics of strains within a single human host (Wolff et al., 2023). This body of work raises the question of whether the SLM can capture patterns of strain level diversity across hosts. Because genetic variants are the constituents of strains, this question can be tested as a question of allelic patterns across hosts and whether the SLM succeeds (Lines 69-72). 

It requires a lot of work for a reader to understand what is being referred to at each point in the text. For example:

1) At Line 94: “the empirical distribution” is unclear; “the distribution of within-host allele frequencies across all hosts” would be clearer. Even clearer would be “the distribution of within-host frequencies at a given site, across all hosts” or something else.

Author response: When referring to a distribution in the revised manuscript I now specify the quantity of a distribution (e.g., Line 106).

2) Line 339: “unlikely for the allele frequency fluctuations I was able to observe”. Does this refer to fluctuations across hosts, or time? At the very least, a figure panel should be referred to here.

Author response: I thank the Reviewer for pointing out this typo. “Allele frequency fluctuations” should be “range of allele frequencies”, as the proceeding sentence discusses the lower bound on allele frequencies that could be inferred after accounting for sequencing error. This typo has been corrected in the revised manuscript. The data referred to in this sentence is summarized in the proceeding sentence (Lines 375-378).

3) Line 418: what is a “true strain absence”. Is the author suggesting that single strain colonization is rare for all species and strains? This would conflict with the literature (e.g. B fragilis almost always colonizes humans as a single strain, PMID: 30673701), but if it is supported by the data this should be made more clear.

Author response: By “true strain absence” I mean strain structure that we do not observe in a host because it is not in the host, as opposed to strain structure that is present at an abundance in a host that is too low to be detected. The gamma distribution implies that the perceived absence of strain structure is typically driven by insufficient depth of sampling (Grilli, 2020), a conclusion that is consistent with the paper cited by the Reviewer (Garud et al., 2019). I have revised this section of the manuscript to clarify these points (Lines 457-471).

4) Line 104: which variable? Mean and variance across what?

Author response: I rescaled allele frequencies by pooling all sites across all hosts and calculating the mean and variance. I rescaled mean allele frequencies by pooling all sites and calculating the mean and variance. I now clarify these points in the revised manuscript (Lines 114-120; 137-138). 

5) Line 500: I’m not sure exactly what point is trying to be made here. If there is already an interpretation of the data (antibiotics wiped out a strain), how does the SLM help?

Author response: The utility of an interpretation depends on the goal of the study. A course of antibiotics is a particularly strong perturbation that would alter the dynamics of strains. Other, arguably less drastic perturbations could be caused by a number of factors (e.g., a host traveling to a different climate, a temporary change in diet, etc.). In these scenarios it would be useful to have an ecological framework to determine whether the dynamics of strains differ from a quantitative null. The gamma distribution obtained from the SLM provides such a null, allowing researchers to determine whether the strain dynamics we observe truly depart from what we would expect if growth rate, environmental noise, and carrying capacity were the only ecologically important factors. For example, the gamma distribution was recently used as a null model to determine whether interactions between strains were sufficiently strong that the assumptions of the SLM were invalid (Goyal et al., 2022). In the revision, I have revised this section of the manuscript to incorporate the Reviewer’s feedback and increase overall clarity (Lines 554-558). 

Technical notes

6) I find Figure 1 (as well as some s to not be presented in a manner such that I can evaluate the claim being made: that the rescaled distributions are similar across species. To evaluate this, I would like to see example distributions (perhaps cumulative curves) drawn for each species – instead what I am presented with is a mess of dots on top of each other in 1b and 1d. I see notable differences, even though these are rescaled already. No goodness of test fits are applied and no alternative distributions are provided.

Author response: The purpose of Fig. 1b is to provide empirical motivation for examining the SLM as a model of strain ecology across hosts. A gamma appears to capture the distribution, so one should investigate models of ecology that generate a gamma distribution. I propose the SLM as such a model. In the revision I have used a standard fitting procedure that I now describe in the manuscript (Lines 120-134). Because the distribution in Fig. 1b uses tens of thousands of data points, a goodness of fit test would inevitably produce a P-value less than 0.05, suggesting that a more insightful approach would be to compare the fit of the gamma with alternative distributions.

There are many options for alternative distributions, though the Reviewer did not recommend a specific one. Since the empirical Abundance Fluctuation Distribution (AFD) in Fig. 1c is wide (stretches over several orders of magnitude), a Gaussian would be inappropriate on the outset. As a reasonable alternative, I chose the lognormal as it has been used in the past as a macroecological comparison for the gamma AFD (Grilli, 2020). By fitting a lognormal, we see that it in no way captures the empirical distribution (Fig. 1c). This result is even more apparent when plotting the same data as a survival distribution, as recommended by the reviewer. Here we see that all species except one follow the gamma distribution (Fig. S2 in the revised manuscript, embedded below). I cannot compare these distributions using a likelihood ratio test since they are not nested, but by calculating the Akaike Information Criterion it is clear that the gamma is a far better descriptor of the empirical distribution (〖AIC〗_gamma= 6,277,330,〖AIC〗_lognorm=6,618,492; Lines 132-134). The AIC is so high because we are summing a large number of data points for each test.

Regarding distributions of mean abundances, I purposefully chose not to propose a mathematical distribution (e.g., Gaussian, gamma, etc.) to explain the empirical distribution in Fig. 1d because a mathematical descriptor of the distribution is not relevant to the subsequent analyses. Using the sampling form of the gamma we can only describe the patterns of genetic diversity across hosts for a given site using the empirical mean and variance. We are not interested in suggesting a probability distribution to capture the empirical distribution of means because the mean is an empirical input used to calculate the predicted value of prevalence (Eq. 6). This point has been clarified in the revised manuscript (Lines 143-152).

Regarding the appropriateness of the gamma, I explicitly test the gamma distribution by using the mean and variance to predict the prevalence of an allele across hosts (Fig. 2-4). This is a prediction obtained using only empirical inputs (i.e., the mean and variance) with zero free parameters. Alternatively stated, no statistical fits were performed for that analysis. 

7) Site frequency spectra are not widely used in microbiology, as the SFS is something that results from recombination. The idea that within-host or across-host evolution would result in a meaningful SFS is confusing to this reviewer.

Author response: The SFS has been intensively examined in theoretical studies motivated by microbial evolution, including studies that examine purely asexually evolving populations (Cvijović et al., 2018; Kosheleva & Desai, 2013; Neher & Hallatschek, 2013; Okada & Hallatschek, 2021) as well as populations evolving with varying rates of recombination (Good et al., 2014; Neher et al., 2013). Furthermore, the SFS has been the primary data object used for strain inference, whether it is by quantifying the existence of strain structure (e.g., StrainFinder in Smillie et al. (2018)) or by obtaining a single haplotype from the SFS using the unique form generated by the existence of co-occurring strains (Garud, Good et al., 2019).

The reviewer is correct in that one’s interpretation of the SFS depends on the degree of recombination in the population. There is increasing evidence that bacteria recombine more often than previously thought. The decay of measures of correlation in genotype frequencies (e.g., linkage disequilibrium) decays with genetic distance (i.e., # base pairs) at a faster rate than expected by an asexual population (Garud, Good et al., 2019; Good, 2022). While this rate of decay is slower than what might be expected under free recombination, it does mean that an asexual view of bacterial evolution, and how that view shapes our understanding of empirical patterns such as the SFS, is not entirely valid. 

However, in all these studies, and in the field at large, the SFS is defined as the distribution of allele frequencies among all sites within a single host/population. In this study I solely refer to the distribution of allele frequencies at a single site across hosts, a distribution that has previously been defined as the single-site frequency spectrum (Theys et al., 2018). 

8) Fig 2c: I’m not sure what the value of predicting “mean within-host allele frequency across hosts” is, if all hosts—including those without the allele at all—are included. Perhaps I am missing something. Also, it seems like most of the data here isn’t fitting the line, but instead fitting a curve starting at y = 10^-1. These graphs should be made as density maps to make it clearer how well the data and the prediction line up.

Author response: In this manuscript I predict the fractions of hosts harboring a given allele (i.e., prevalence). I do not predict the mean frequency of an allele across hosts, that quantity is an empirical input used to calculate the predicted prevalence (Eq. 6). Regarding why we want to include hosts where an allele was not observed in our calculation of the mean, we are hypothesizing that allele frequencies across hosts can be described using a gamma distribution, the stationary solution of the SLM. If the underlying empirical data truly followed the form of the gamma distribution I used that explicitly accounts for total depth of coverage (i.e., sampling; Eq. 4), then excluding hosts where an allele was not observed (i.e., the allele had a coverage of zero) means that I would no longer be sampling from that distribution, I would instead be sampling from a truncated form of that distribution. 

Regarding the data, first, the black dashed line is a prediction using measures obtained from empirical data (i.e., the mean and variance) with zero free parameters. It is not a statistical fit because there were no free parameters to infer. I now clarify this point throughout the manuscript, including at lines that cite Fig. 2c (Lines 77-80; 277-279; 657-660), and the figure legend.

Second, I stated in the original submission that the prediction tended to succeed for alleles with high mean frequency. We expect that some fraction of polymorphic sites for a given species within a host are polymorphic because they are on the background of a strain (i.e., ecology), whereas other sites are segregating due to evolution. This result is consistent with the conceptual picture that both ecology and evolution are operating within a given species, but it is novel in that I was able to predict the prevalence of alleles with a high mean abundance across hosts using an ecological model that succeeded at predicting the dynamics of strains within hosts (Wolff et al., 2023). I now clarify these points in the revised manuscript (Lines 441-456).

The figures presented in the original manuscript were already illustrated as density maps. The color of each data point is proportional to the density of points in its immediate area. However, this detail was not made explicit, so the revised manuscript now explicitly mentions this detail in the figure legend. There was a plotting error in the original manuscript, as the color did not correspond to the density. This has been corrected in the revised manuscript. I have also added a color bar indicating how the color corresponds to the number of points for all figures with density plots (Figs. 2, S5, S6, S9, S10, S12, S13).

Some thoughts: It is clear that the author has put a lot of work into the generation of this data and model, but it is always important to be clear about what is known and what is novel. The formalization of an already proposed model can be valuable. Presenting a model which more clearly shows what has been shown by others could also be valuable; if this is the goal than more effort needs to be put into making it clear.

Author response: I agree with the reviewer. The novelty of this study is that I demonstrated how an ecological model that captures the dynamics of strains within a single host (Wolff et al., 2023) was able to capture patterns of genetic diversity across hosts due to the existence of strain structure. In the revision I have incorporated their feedback by carefully clarifying in the Abstract, Introduction, and Discussion the novelty of this work and how it builds on prior studies (Lines 56-72; 73-83; 441-456).

Reviewer #2

Major comments

Generally, this is an interesting work. However, as someone with a limited background in biostatistics, I find the description of the underlying modeling approach very hard to follow. For instance, the assumptions behind the SLM described from line 197-208 are not clear to me. Hence, my recommendation is to clarify the description of the SLM, its assumptions, and the overall discussed concepts for readers less familiar with them. I understand that this is not my area of expertise, but it would be very helpful to make the presented results more accessible to scientists in the gut microbiome field.

Author response: I thank the reviewer for their comments. I have worked to clarify the motivation for and definition of the SLM in the revised manuscript. I have added brackets in Eq. 1 to describe the physical meaning of each term in the equation and additional text to describe how noise is modeled in the equation (Lines 211-242). Additional technical steps are provided in the Materials and Methods (Lines 633-679). As for the rest of the manuscript, I have revised several sections in the Abstract, Introduction, and Discussion to clarify novelty of this work and how it builds on prior studies (Lines 56-72; 73-83; 441-456).

Minor comments 

1) The legends for Figures 1-4 are more a repetition of the Results section than a clear description of what is shown in each panel.

Author response: I initially included additional descriptive details in the legends at the request of past reviewers. The legends of Fig. 1-4 have now been revised to emphasize the contents of the figures rather than the results. 

2) Line 339: MAGPD is mentioned here for the first time without having been defined before.

Author response: A definition of the acronym has now been included and moved earlier in the manuscript to where data from MAPGD was first described (Line 97).

3) Line 448-449: “Regardless of the specific model, the conclusion that a substantial fraction of observable alleles are primarily subject to ecological, as opposed to evolutionary, dynamics is of consequence to studies of genetic diversity in the human gut.” It is not clear to me which part of the Results section precisely shows this.

Author response: In this manuscript I demonstrate that the Stochastic Logistic Model of growth, an ecological model that has been used to capture the temporal dynamics of strains within a single host, can predict patterns of allelic prevalence across hosts for alleles of moderate-to-high mean frequency (Fig. 2, 3). Because alleles are the constituents of strains, this result means that the across-host patterns of a substantial portion of alleles are driven by strain-level ecology. This result is supported by the observation that the error of the prevalence predictions is inversely related to the fraction of hosts harboring strains (Fig. 4). I have revised the Discussion to clarify this point (Lines 441-456). 

4) Line 655: There is a figure legend for Figure 5, which does not exist.

Thank you for pointing this out. The figure legend has been removed and there never was an intention of having a fifth figure. It was an artifact of reformatting the paper in multiple latex templates. 

5) Line 704: Is this really the correct URL for the Human Microbiome Project?

Author response: I have used a URL that links to a repository containing the data of the Human Microbiome Project. This URL and the data therein were listed under the Data and Code Availability Statement of a previously published paper that also used Human Microbiome Project data (Garud, Good, et al., 2019). In the revision I now include the official URL of the Human Microbiome Project alongside this link.

References

Cvijović, I., Good, B. H., & Desai, M. M. (2018). The Effect of Strong Purifying Selection on Genetic Diversity. Genetics, 209(4), 1235–1278. https://doi.org/10.1534/genetics.118.301058

Garud, N. R., Good, B. H., Hallatschek, O., & Pollard, K. S. (2019). Evolutionary dynamics of bacteria in the gut microbiome within and across hosts. PLOS Biology, 17(1), e3000102. https://doi.org/10.1371/journal.pbio.3000102

Good, B. H. (2022). Linkage disequilibrium between rare mutations. Genetics, 220(4), iyac004. https://doi.org/10.1093/genetics/iyac004

Good, B. H., Walczak, A. M., Neher, R. A., & Desai, M. M. (2014). Genetic Diversity in the Interference Selection Limit. PLOS Genetics, 10(3), e1004222. https://doi.org/10.1371/journal.pgen.1004222

Goyal, A., Bittleston, L. S., Leventhal, G. E., Lu, L., & Cordero, O. X. (2022). Interactions between strains govern the eco-evolutionary dynamics of microbial communities. ELife, 11, e74987. https://doi.org/10.7554/eLife.74987

Grilli, J. (2020). Macroecological laws describe variation and diversity in microbial communities. Nature Communications, 11(1), 4743. https://doi.org/10.1038/s41467-020-18529-y

Kosheleva, K., & Desai, M. M. (2013). The Dynamics of Genetic Draft in Rapidly Adapting Populations. Genetics, 195(3), 1007–1025. https://doi.org/10.1534/genetics.113.156430

Neher, R. A., & Hallatschek, O. (2013). Genealogies of rapidly adapting populations. Proceedings of the National Academy of Sciences, 110(2), 437–442. https://doi.org/10.1073/pnas.1213113110

Neher, R. A., Kessinger, T. A., & Shraiman, B. I. (2013). Coalescence and genetic diversity in sexual populations under selection. Proceedings of the National Academy of Sciences, 110(39), 15836–15841. https://doi.org/10.1073/pnas.1309697110

Okada, T., & Hallatschek, O. (2021). Dynamic sampling bias and overdispersion induced by skewed offspring distributions. Genetics, 219(4), iyab135. https://doi.org/10.1093/genetics/iyab135

Roodgar, M., Good, B. H., Garud, N. R., Martis, S., Avula, M., Zhou, W., Lancaster, S. M., Lee, H., Babveyh, A., Nesamoney, S., Pollard, K. S., & Snyder, M. P. (2021). Longitudinal linked-read sequencing reveals ecological and evolutionary responses of a human gut microbiome during antibiotic treatment. Genome Research, 31(8), 1433–1446. https://doi.org/10.1101/gr.265058.120

Smillie, C. S., Sauk, J., Gevers, D., Friedman, J., Sung, J., Youngster, I., Hohmann, E. L., Staley, C., Khoruts, A., Sadowsky, M. J., Allegretti, J. R., Smith, M. B., Xavier, R. J., & Alm, E. J. (2018). Strain Tracking Reveals the Determinants of Bacterial Engraftment in the Human Gut Following Fecal Microbiota Transplantation. Cell Host & Microbe, 23(2), 229-240.e5. https://doi.org/10.1016/j.chom.2018.01.003

Theys, K., Feder, A. F., Gelbart, M., Hartl, M., Stern, A., & Pennings, P. S. (2018). Within-patient mutation frequencies reveal fitness costs of CpG dinucleotides and drastic amino acid changes in HIV. PLOS Genetics, 14(6), e1007420. https://doi.org/10.1371/journal.pgen.1007420

Wolff, R., Shoemaker, W., & Garud, N. (2023). Ecological Stability Emerges at the Level of Strains in the Human Gut Microbiome. MBio, 0(0), e02502-22. https://doi.org/10.1128/mbio.02502-22

---

## [Decision Letter · Decision Letter 1]

22 May 2023

PONE-D-22-30239R1A macroecological perspective on genetic diversity in the human gut microbiomePLOS ONE

Dear Dr. Shoemaker,

Thank you for submitting your manuscript to PLOS ONE. After careful consideration, we feel that it has merit but does not fully meet PLOS ONE’s publication criteria as it currently stands. Therefore, we invite you to submit a revised version of the manuscript that addresses the points raised during the review process.

We look forward to receiving your revised manuscript.

Kind regards,

Karthik Raman, Ph.D.

Academic Editor

PLOS ONE

Additional Editor Comments:

I am in agreement with one of the reviewers, who has given detailed constructive criticism to further improve the manuscript. It is important that the manuscript be carefully revisited to systematically address all the concerns of the reviewers.

Reviewers' comments:

Reviewer's Responses to Questions

**Comments to the Author**

1. If the authors have adequately addressed your comments raised in a previous round of review and you feel that this manuscript is now acceptable for publication, you may indicate that here to bypass the “Comments to the Author” section, enter your conflict of interest statement in the “Confidential to Editor” section, and submit your "Accept" recommendation.

Reviewer #1: (No Response)

Reviewer #2: All comments have been addressed

2. Is the manuscript technically sound, and do the data support the conclusions?

Reviewer #1: Partly

Reviewer #2: Yes

3. Has the statistical analysis been performed appropriately and rigorously? 

Reviewer #1: I Don't Know

Reviewer #2: (No Response)

4. Have the authors made all data underlying the findings in their manuscript fully available?

Reviewer #1: Yes

Reviewer #2: Yes

5. Is the manuscript presented in an intelligible fashion and written in standard English?

Reviewer #1: No

Reviewer #2: Yes

6. Review Comments to the Author

Reviewer #1: This manuscript has improved from the last version. In R2R document, the author states that the purpose of this manuscript is to demonstrate that the previously proposed model, in which (1) most intrastrain polymorphisms arise from migration and growth of strains rather than within-person evolution; and (2) this is formalized with an SLM model, can explain allelic prevalence across hosts. Bolstering the SLM model and the importance of strain structure is a fine goal (though I have a question on the contextualization of this below), and now I can begin to interpret the manuscript.

Significant work still remains to be done on the writing side. While the author appears to have attempted to address my concerns, shortcuts were taken rather than addressing the real problem (e.g. point #1f below), the text still falls short of communicating what is being advanced by these models, and murky language is used throughout--- such that I still relied heavily on the response to the reviewer (R2R) to interpret what the author intended. I do not think that all interdisciplinary work needs to be so difficult to read. I also have major concerns around the framing and some technical aspects of this work.

1) The writing is murky and it is hard to follow what point the author is making throughout.

1a) As an example to the author, I provide below an edited alternative of the middle of the abstract which is easier to read. In this example, the connection between strains and ecology is made more clear to a naiive reader, and what the SLM is modeling (strains) is made clear within the sentence that introduces it. The use of a “however” is removed from a location where a clear contrast isn’t being made. The phrases “determine whether” and “is capable of capturing” are replaced by more direct language. I am not sure if the suggestion for the last sentence reflects what the author intended; if I am incorrect, the author should clarify what is meant by “patterns of genetic diversity across hosts follow statistically similar forms”.

Revised abstract:

“…Recent efforts have suggested that a large fraction intrahost, intraspecies, genetic variation is driven by dynamics between co-colonizing of strains of the same species, highlighting the importance of modeling ecological forces. In particular, the Stochastic Logistic Model (SLM) of growth, commonly used in macroecology to describe between-species variation, has been used successfully to predict the temporal dynamics of strains within a single human host. Here, using data from 22 common microbial species across a large cohort of unrelated hosts, I show that the SLM also successfully predicts across-host genetic diversity in the human gut. The SLM predicts both the distribution of allele frequencies across hosts and the fraction of hosts harboring an allele for a given site (i.e., prevalence). The accuracy of the SLM in predicting these across-host parameters is correlated with independent estimates of strain structure, confirming that the success of the SLM arises from the presence of strain-level ecology in the human gut.

1b) In regards to my point #2 in the previous round the authors have changed a phrase but have not fixed the core of the problem, which is requiring a reader to read every sentence in order and guess what is being referred to be able to follow. One should not write “allele frequency fluctuations that I was able to observe” when they could say something like “THE LARGE allele frequency fluctuations ACROSS HOSTS” or whatever feature the author is actually intending here. As a reader, I cannot evaluate the validity of a sentence that is vague.

1c) Line 62 would be clearer if the manuscript said “In macroecology, the SLM…. “

1d) The last paragraph of the introduction would be much clearer with specific methods (is “an ecological model” different than the SLM already discussed? In the last sentence, what is “an alternative evolutionary model”?) or with the general methods removed (just state result).

1e) “First, I examined the distribution of within-host allele frequencies across hosts.” This should be explained better. This is not a normal metric, and it would be great if this could be spelled out.

1f) Line 70: “If the typical allele was present due to evolutionary forces, then the empirical distribution of within-host allele frequencies across hosts can be viewed as an ensemble of single-site frequency spectra.” By definition I believe that this metric is an ensemble of SFS. I think you mean “… across hosts WOULD BE EQUIVALNT TO THE ensemble of single-site frequency spectra EXPECTED FROM WITHIN PERSON EVOLUTION.” It would also be helpful for the reader if the shape of each of these expectations were described in terms of the distributions used in Figure 1 in this paragraph.

1g) The dN/dS curve mentioned is all for Bacteroides and related species. This curve has not been investigated in other major taxa (e.g. Clostridia). In addition, this result is mentioned but not explained well enough for a reader to understand this point. “a clear example being the observation 41 that THE RATIO OF NONSYNONYMOUS DIVERGENCE TO SYNONYMOUS DIVERGENCE DECAYS NEARLY IDENTICALLY OVER LONG TIME SCALES OF across microbial species in the human gut”

1h) “This disproportionate focus on 32 individual species and differences between species can lead to idiosyncratic notions 33 about the typical dynamics of a genetic variant in the microbiome….. Instead, it is reasonable to start by identifying genetic patterns” I am just not sure what this means. One interpretation of these sentences is that the author is claiming that looking at things one species at a time is incorrect. I’m not sure the author needs to say this. I think it would be enough to say that identifying a single model that works across species would reveal general principles of eco-evo dynamics across microbiomes.

There are parts of the results section that are likewise confusing, but I have selected to use my time to give several clear examples of areas for improvement in the introduction and results as an example to the author of how more clarity can be achieved.

Other major concerns:

2) It is very true that new and better eco-evo models are needed for microbial populations, but the presentation of current knowledge in the population genetics field and the remaining gaps are incorrectly summarized.

2a) The surprise of strain structure is a bit overstated. One can infer that strains are a real entity by building phylogenetic trees from available isolates or from metagenomic samples with single strains (e.g. quasiphasable as in Garud and Good et al). This sort of expectation should be more explicitly stated in the introduction.

2b) Page 2, Line 20: “Such dynamics are a clear departure from those captured by standard population genetic models, where genetic variants either arise in a population due to mutation or are introduced by migration and then proceed 22 towards extinction or fixation (i.e., origin-fixation models), suggesting that measures of genetic diversity estimated within the human gut are shaped by the ecology of strains alongside evolution [10]. “ There are many population genetic models that deal with standing variation – the entire field of human population genetics on quantitative traits comes to mind. I think the bigger gap is that models for population genetics on standing variation don’t deal with the genome-wide linkage inherent to bacteria.

2c) “This confluence of ecological and evolutionary dynamics calls into question the 26 feasibility of characterizing genetic diversity in the human gut.” “Calls into question the feasibility” is a bit overstated. How about “requires new approaches and theory for”

3) I am not swayed by the argument that there are always multiple strains in a subject (response to previous point #3). B fragilis is often found at very high abundances, and the presences of strain structure does not correlate with the abundance in that species, as would be expected if there was a detection limit problem. The author has not proved “a lack of genuine absences of strain structure” which is very strong claim that when not written in the double negative. This could be mitigated by putting a qualifier of “for most species” for most sentences in this paragraph.

4) If I am understanding correctly, this paper is about the cross-host applicability of the SLM. However, the examples in the last paragraph of the discussion are all about within-host dynamics, which were investigated in a prior manuscript. This is confusing and perhaps misleading. Could the across-host applicability of the SLM help us understand how the microbiome at a global scale might change in response to a global perturbation like global warming?

5) I understand that Figure 1 isn’t about a goodness of fit, though the author has now done some analysis in this direction now anyway. I still don’t really understand what I am supposed to learn from this data looking roughly gamma though. How do the lognormal and gamma distribution relate to the SFS expected for within-host evolution described in lead up to Figure 1? Can the author give an example of a generative process that wouldn’t result in the patterns shown in Figure 1?

Reviewer #2: In their revision, the author worked to improve the readability of the manuscript, clarifying the concepts and the aim of the study. Overall, they did a good job. I thank the author for rewriting the manuscript to make it easier to follow. KI still have some difficulties interpreting the equations, which is likely due to my unfamiliarity with the subject.

Overall comments:

- Since the article was based on a previously published work (Wolff et al. 2023), I think it would improve understanding of the context of the study further if one or two lines summarizing the results of that study were added at line 69.

- While the last paragraph of the discussion addresses this, it is still not entirely clear to me how the model can be applied to dysbiosis. What are the implications of the prediction of the prevalence of a given gene in a fraction of hosts? What does that tell us about the microbiome?

7. PLOS authors have the option to publish the peer review history of their article (what does this mean?). If published, this will include your full peer review and any attached files.

Reviewer #1: No

Reviewer #2: No

---

## [Author Response · Author response to Decision Letter 1]

8 Jun 2023

Editor Comments

I am in agreement with one of the reviewers, who has given detailed constructive criticism to further improve the manuscript. It is important that the manuscript be carefully revisited to systematically address all the concerns of the reviewers.

Author response: I thank the editor for the feedback along with the opportunity to improve the manuscript. In my response I describe how all comments made by both reviewers were addressed in my resubmission. 

Reviewer #1

This manuscript has improved from the last version. In R2R document, the author states that the purpose of this manuscript is to demonstrate that the previously proposed model, in which (1) most intrastrain polymorphisms arise from migration and growth of strains rather than within-person evolution; and (2) this is formalized with an SLM model, can explain allelic prevalence across hosts. Bolstering the SLM model and the importance of strain structure is a fine goal (though I have a question on the contextualization of this below), and now I can begin to interpret the manuscript.

Author response: I thank the reviewer for their constructive comments before, which have improved the manuscript.

Significant work still remains to be done on the writing side. While the author appears to have attempted to address my concerns, shortcuts were taken rather than addressing the real problem (e.g. point #1f below), the text still falls short of communicating what is being advanced by these models, and murky language is used throughout--- such that I still relied heavily on the response to the reviewer (R2R) to interpret what the author intended. I do not think that all interdisciplinary work needs to be so difficult to read. I also have major concerns around the framing and some technical aspects of this work.

1) The writing is murky and it is hard to follow what point the author is making throughout.

Author response: In this revision I have gone to considerable efforts to clarify my points. In addition to incorporating the reviewer’s suggestions, I have gone through the manuscript and identified additional opportunities for clarification. 

1a) As an example to the author, I provide below an edited alternative of the middle of the abstract which is easier to read. In this example, the connection between strains and ecology is made more clear to a naiive reader, and what the SLM is modeling (strains) is made clear within the sentence that introduces it. The use of a “however” is removed from a location where a clear contrast isn’t being made. The phrases “determine whether” and “is capable of capturing” are replaced by more direct language. I am not sure if the suggestion for the last sentence reflects what the author intended; if I am incorrect, the author should clarify what is meant by “patterns of genetic diversity across hosts follow statistically similar forms”.

Revised abstract:

“…Recent efforts have suggested that a large fraction intrahost, intraspecies, genetic variation is driven by dynamics between co-colonizing of strains of the same species, highlighting the importance of modeling ecological forces. In particular, the Stochastic Logistic Model (SLM) of growth, commonly used in macroecology to describe between-species variation, has been used successfully to predict the temporal dynamics of strains within a single human host. Here, using data from common microbial species across a large cohort of unrelated hosts, I show that the SLM also successfully predicts across-host genetic diversity in the human gut. The SLM predicts both the distribution of allele frequencies across hosts and the fraction of hosts harboring an allele for a given site (i.e., prevalence). The accuracy of the SLM in predicting these across-host parameters is correlated with independent estimates of strain structure, confirming that the success of the SLM arises from the presence of strain-level ecology in the human gut.

Author response: I appreciate the reviewer’s suggestions. I have revised the Abstract to incorporate all suggested edits made by the reviewer.

1b) In regards to my point #2 in the previous round the authors have changed a phrase but have not fixed the core of the problem, which is requiring a reader to read every sentence in order and guess what is being referred to be able to follow. One should not write “allele frequency fluctuations that I was able to observe” when they could say something like “THE LARGE allele frequency fluctuations ACROSS HOSTS” or whatever feature the author is actually intending here. As a reader, I cannot evaluate the validity of a sentence that is vague.

Author response: In the prior round of comments the reviewer quoted a line of the manuscript that contained a typo. I fixed the typo and I thank the reviewer for clarifying their prior comment. The revision now specifies that “fluctuations” refers to “fluctuations across hosts” (line 410). I have also gone through the manuscript and clarified every mention of the term “fluctuation”. 

1c) Line 62 would be clearer if the manuscript said “In macroecology, the SLM…. “

Author response: I have made the suggested edit (line 69).

1d) The last paragraph of the introduction would be much clearer with specific methods (is “an ecological model” different than the SLM already discussed? In the last sentence, what is “an alternative evolutionary model”?) or with the general methods removed (just state result).

Author response: I have changed “an ecological model” to “the SLM as a model of ecology” (line 84). I have rewritten the last sentence to clarify that I am referring to evolutionary Langevin equations that also generate the same stationary probability distribution as the SLM (lines 91-93). I then specify that I inferred the presence of strain structure and found that strain structure was correlated with the accuracy of the SLM in predicting allelic prevalence (lines 93-97). 

1e) “First, I examined the distribution of within-host allele frequencies across hosts.” This should be explained better. This is not a normal metric, and it would be great if this could be spelled out.

Author response: I have made the requested edits, changing the quoted line to: 

“First, I obtained the distribution of across-host allele frequencies for each nucleotide site and then pooled the frequencies of all sites.” (line 118-119).

1f) Line 70: “If the typical allele was present due to evolutionary forces, then the empirical distribution of within-host allele frequencies across hosts can be viewed as an ensemble of single-site frequency spectra.” By definition I believe that this metric is an ensemble of SFS. I think you mean “… across hosts WOULD BE EQUIVALNT TO THE ensemble of single-site frequency spectra EXPECTED FROM WITHIN PERSON EVOLUTION.” It would also be helpful for the reader if the shape of each of these expectations were described in terms of the distributions used in Figure 1 in this paragraph.

Author response: I appreciate the reviewer’s comments regarding the importance of clarity. I have incorporated their request into the manuscript (lines 119-123). 

The purpose of Fig. 1c is to illustrate how a single probability distribution is capable of explaining the empirical distributions of a many species. Visual inspection (and AIC tests based on the reviewer’s previous requests) suggests that the gamma distribution sufficiently captures the empirical distribution. I then identify the SLM as a reasonable model because 1) of its past success in explaining within-host strain dynamics and 2) because its stationary distribution is the gamma distribution. I then proceed with testing the gamma at each site (Figs. 2,3) and determine that the accuracy of the gamma is correlated with an independent estimate of strain structure (Fig. 4), validating the ecological interpretation of the gamma distribution. In the revision I identified sections of the manuscript related to the above points that required additional clarity and made appropriate edits (lines 219-227, 253-255). 

1g) The dN/dS curve mentioned is all for Bacteroides and related species. This curve has not been investigated in other major taxa (e.g. Clostridia). In addition, this result is mentioned but not explained well enough for a reader to understand this point. “a clear example being the observation that THE RATIO OF NONSYNONYMOUS DIVERGENCE TO SYNONYMOUS DIVERGENCE DECAYS NEARLY IDENTICALLY OVER LONG TIME SCALES OF across microbial species in the human gut”

Author response: The curve presented in Fig. 3 of Garud, Good et al. was expanded with additional species in Shoemaker et al., where it visualized species from 20 genera, 14 families, 7 orders, 6 classes, and 5 phyla (2019; 2021). The assemblage of species represented in Shoemaker et al. includes members of the class Clostridia (e.g., Clostridium, Butyrivibrio; 2021). I understand that it is not an exhaustive representation of bacterial diversity, but it provides empirical motivation for identifying patterns that hold across evolutionarily distant species in the human gut microbiome. Therefore, I briefly summarize the taxonomic diversity in the line quoted by the reviewer in the revision. I have also edited the line to state that the ratio dN/dS decays with increasing dS, where dS is interpreted as a proxy for evolutionary time (lines 47-51). 

1h) “This disproportionate focus on individual species and differences between species can lead to idiosyncratic notions about the typical dynamics of a genetic variant in the microbiome….. Instead, it is reasonable to start by identifying genetic patterns” I am just not sure what this means. One interpretation of these sentences is that the author is claiming that looking at things one species at a time is incorrect. I’m not sure the author needs to say this. I think it would be enough to say that identifying a single model that works across species would reveal general principles of eco-evo dynamics across microbiomes.

Author response: Fig. 1c-f contains visual examples of what is meant by the line quoted by the reviewer. I believe that it is reasonable to first inspect empirical patterns before applying a model. From past experiences, discussions with peers, and assessing published literature, the practice of plotting distributions of the same quantity from different species/systems on the same axis to identify qualitatively similar behavior is often overlooked in the life sciences. This observation was the motivation for describing the benefits of performing a “data collapse” and investigating whether distributions of within-host allele frequencies for different species have invariant forms (lines 101-107; Fig. 1c). However, the approach that I, and other researchers in microbial evolution and ecology, took is not the singular approach. Rather, it can be viewed as one way to address scientific questions rather than a prescription and I have revised the manuscript to reflect this view and the views of the reviewer (lines 38-45). 

There are parts of the results section that are likewise confusing, but I have selected to use my time to give several clear examples of areas for improvement in the introduction and results as an example to the author of how more clarity can be achieved. 

Author response: I have combed through the manuscript and done my best to meet their requests. I appreciate the reviewer for identifying areas where clarity can be improved. 

Other major concerns: 

2) It is very true that new and better eco-evo models are needed for microbial populations, but the presentation of current knowledge in the population genetics field and the remaining gaps are incorrectly summarized.

2a) The surprise of strain structure is a bit overstated. One can infer that strains are a real entity by building phylogenetic trees from available isolates or from metagenomic samples with single strains (e.g., quasiphasable as in Garud and Good et al). This sort of expectation should be more explicitly stated in the introduction.

Author response: In this manuscript I start from the observation that a number of genetic variants at the species level neither become fixed nor go extinct over extended periods of time within a single human gut. This empirical motivation has been used in other manuscripts to introduce readers to the concept of multiple strains co-occurring within a single human host (Good & Hallatschek, 2018). 

In lines 11-16 I altered the language to be more descriptive so that I do not overstate the existence of strain structure. At the request of the reviewer, I have included a sentence discussing how this ecological structure is often reflected in the shape of phylogenetic trees constructed from microbial isolates (lines 13-16).

2b) Page 2, Line 20: “Such dynamics are a clear departure from those captured by standard population genetic models, where genetic variants either arise in a population due to mutation or are introduced by migration and then proceed towards extinction or fixation (i.e., origin-fixation models), suggesting that measures of genetic diversity estimated within the human gut are shaped by the ecology of strains alongside evolution [10]. “ There are many population genetic models that deal with standing variation – the entire field of human population genetics on quantitative traits comes to mind. I think the bigger gap is that models for population genetics on standing variation don’t deal with the genome-wide linkage inherent to bacteria.

Author response: The existence of strain structure in microbial communities is not a matter of standing genetic variation in the way it is for humans. Microbial strains that are present the same host represent a form of ecological structure. Namely, what we observe as standing genetic variation that persists over extended timescales (over a year in Wolff et al., 2023) is in reality the presence of multiple community members that occupy different ecological roles while de novo mutations continue to be acquired and segregate within each strain (Dapa et al., 2023; Roodgar et al., 2021). 

This empirical observation has motivated those in the field to develop models that integrate principles from microbial ecology with those from population genetics. For example, in Good, Martis, et al. the authors develop an evolutionary model where each strain acquires mutations that impact growth rate (i.e., fitness) at one rate and mutations that impact which resources they consume at another rate (i.e., ecological strategy; 2018). This model captures the emergence and coexistence of strains within a human host over time, where each individual strain can be viewed as an asexual population where the population scaled mutation rate (population size multiplied by the per-genome mutation rate) is sufficiently small such that typically only a single mutation is segregating within a given strain (strong selection weak mutation limit). Subsequent modeling efforts extend this model to the genome-wide linkage regime where asexually reproducing strains have a population scaled mutation rate sufficiently high so that multiple mutations segregate within a single strain (Wong and Good, 2022).

Regarding recombination, there is no doubt that linkage contributes to the patterns of genetic diversity we observe in the human gut as this topic remains an active area of research (e.g., Garud, Good, 2019; Roodgar et al., 2021; Good, 2022). However, the extent that an absence of recombination is necessary to preserve the ecological structure of strains within a human host is currently unknown, as published eco-evolutionary models of strain dynamics have primarily examined purely asexually reproducing populations. To reflect this point, in the revision I incorporated the reviewer’s comment regarding the need to consider genetic linkage (line 29-31).

2c) “This confluence of ecological and evolutionary dynamics calls into question the feasibility of characterizing genetic diversity in the human gut.” “Calls into question the feasibility” is a bit overstated. How about “requires new approaches and theory for

Author response: I have made the suggested edit (lines 32-33).

3) I am not swayed by the argument that there are always multiple strains in a subject (response to previous point #3). B fragilis is often found at very high abundances, and the presences of strain structure does not correlate with the abundance in that species, as would be expected if there was a detection limit problem. The author has not proved “a lack of genuine absences of strain structure” which is very strong claim that when not written in the double negative. This could be mitigated by putting a qualifier of “for most species” for most sentences in this paragraph.

Author response: The reviewer is correct that the implications of the gamma with regards to the existence of strains makes strong claims. In this manuscript I do not attempt to prove nor disprove this claim, I simply state the interpretation of the gamma based on past macroecological research performed at the species level (Grilli, 2020). In the revision I now clarify this point (lines 493-501). I have also incorporated the reviewer’s suggestion and now include the qualifier “for several species” throughout the paragraph (lines 357, 495, 500, 505). 

4) If I am understanding correctly, this paper is about the cross-host applicability of the SLM. However, the examples in the last paragraph of the discussion are all about within-host dynamics, which were investigated in a prior manuscript. This is confusing and perhaps misleading. Could the across-host applicability of the SLM help us understand how the microbiome at a global scale might change in response to a global perturbation like global warming?

Author response: I apologize for the lack of clarity. The last paragraph was originally written at the request of a previous reviewer asking that I talk about the applicability of the SLM in general. To incorporate this comment and the comment of Reviewer 2, I have rewritten this paragraph to focus on the overall state of strain macroecology in-light of this study and how the assumption of stationarity could be leveraged to identify the effects of perturbations across-hosts (lines 567-587).

The reviewer does raise an interesting question regarding global perturbations. I would imagine that the temperature of endothermic systems like the human gut would be largely immune to slight (though globally important) gradual changes in the temperature of the climate. Scenarios where the gut ecology of a large number of human hosts are simultaneously perturbed would be ideal for evaluating across-host strain ecology. This scenario is most likely to occur in human experimental trials (e.g., effect of change in diet, novel drug testing, etc.). In this scenario the SLM could be used to examine statistical measures of over an ensemble of hosts (e.g., mean abundance over hosts of a given strain) relaxes over time towards a stationary state after a perturbation (i.e., 〈x|t,x_0〉 => 〈x〉). 

5) I understand that Figure 1 isn’t about a goodness of fit, though the author has now done some analysis in this direction now anyway. I still don’t really understand what I am supposed to learn from this data looking roughly gamma though. How do the lognormal and gamma distribution relate to the SFS expected for within-host evolution described in lead up to Figure 1? Can the author give an example of a generative process that wouldn’t result in the patterns shown in Figure 1?

Author response: In my prior revision I fit a lognormal distribution at the reviewer’s request that I include and compare an additional distribution as an alternative to the gamma. The purpose of Fig. 1c is to 

1) Demonstrate that different species have similar distributions of within-host allele frequencies across hosts when rescaled and 

2) Propose the gamma distribution as a potential model of allelic diversity across hosts. 

This is the empirical motivation that leads to the SLM being identified as a reasonable model, as the stationary solution of the SLM predicts a gamma distribution and describes the temporal dynamics of strains within a single human host (Eqs. 10-12). In contrast, a Langevin equation that models linear ecological growth (as opposed to the logistic growth of the SLM) and environmental noise (the same form of noise as the SLM) predicts a lognormal distribution. This model would be inappropriate since the lognormal did a comparatively poor job explaining the empirical distribution in Fig.1c relative to the gamma distribution. 

However, different Langevin equations can produce the same probability distribution and it is worth considering whether such equations are viable alternatives to the SLM. In the manuscript I identified two Langevin equations of molecular evolution that also predict a gamma distribution (Eqs. 7-9; S1 Text). I describe the features of these models, their parameters, and how the assumptions necessary to explain the existence of genetic variants that are present at intermediate frequencies (0 < f < 1) over a large number of hosts are unrealistic (lines 409-439). As for a generative process that would not produce the pattern shown in Fig. 1c and would not produce a lognormal, an evolutionarily neutral allele or an allele on the background of an ecologically neutral strain would generate a Gaussian distribution over short timescales. 

Finally, this section leads into an explicit test of whether the SLM as an ecological interpretation of the gamma is reasonable, where I find that the accuracy of the gamma is correlated with independent estimates of strain structure (Fig. 4), pointing to the SLM as a feasible ecological model. I have revised this section of the manuscript for clarity (Lines 367-386). I now also describe how distributions different from the gamma (i.e., lognormal or Gaussian) could emerge as a consequence of ecological or evolutionary dynamics (lines 373-385). 

Reviewer #2

In their revision, the author worked to improve the readability of the manuscript, clarifying the concepts and the aim of the study. Overall, they did a good job. I thank the author for rewriting the manuscript to make it easier to follow. I still have some difficulties interpreting the equations, which is likely due to my unfamiliarity with the subject.

Author response: I thank the reviewer for their feedback.

Overall comments

Since the article was based on a previously published work (Wolff et al. 2023), I think it would improve understanding of the context of the study further if one or two lines summarizing the results of that study were added at line 69.

Author response: I have added the requested lines to the revision (lines 76-80).

While the last paragraph of the discussion addresses this, it is still not entirely clear to me how the model can be applied to dysbiosis. What are the implications of the prediction of the prevalence of a given gene in a fraction of hosts? What does that tell us about the microbiome?

Author response: I apologize to the reviewer for the lack of clarity. Much of this paragraph was written at the request of a past reviewer that asked for the macroecological implications of this work in terms of practical application to be discussed. To incorporate this comment and the comment of Reviewer 1, I have rewritten this paragraph to focus on the state of strain macroecology in-light of this study. Specifically, how the analyses in this study assume that strain dynamics are stationary with respect to time and how that assumption could be leveraged to determine when statistics calculated over the gut microbiomes of a large number of perturbed hosts approach stationarity (e.g., mean frequency or prevalence approaching a stationary value over time; lines 576-587). 

References

Dapa, T., Wong, D. P., Vasquez, K. S., Xavier, K. B., Huang, K. C., & Good, B. H. (2023). Within-host evolution of the gut microbiome. Current Opinion in Microbiology, 71, 102258. https://doi.org/10.1016/j.mib.2022.102258

Garud, N. R., Good, B. H., Hallatschek, O., & Pollard, K. S. (2019). Evolutionary dynamics of bacteria in the gut microbiome within and across hosts. PLOS Biology, 17(1), e3000102. https://doi.org/10.1371/journal.pbio.3000102

Good, B. H. (2022). Linkage disequilibrium between rare mutations. Genetics, 220(4), iyac004. https://doi.org/10.1093/genetics/iyac004

Good, B. H., Martis, S., & Hallatschek, O. (2018). Adaptation limits ecological diversification and promotes ecological tinkering during the competition for substitutable resources. Proceedings of the National Academy of Sciences, 115(44), E10407–E10416. https://doi.org/10.1073/pnas.1807530115

Good, B. H., & Hallatschek, O. (2018). Effective models and the search for quantitative principles in microbial evolution. Current Opinion in Microbiology, 45, 203–212. https://doi.org/10.1016/j.mib.2018.11.005

Grilli, J. (2020). Macroecological laws describe variation and diversity in microbial 

communities. Nature Communications, 11(1), 4743. https://doi.org/10.1038/s41467-020-18529-y

Roodgar, M., Good, B. H., Garud, N. R., Martis, S., Avula, M., Zhou, W., Lancaster, S. M., Lee, H., Babveyh, A., Nesamoney, S., Pollard, K. S., & Snyder, M. P. (2021). Longitudinal linked-read sequencing reveals ecological and evolutionary responses of a human gut microbiome during antibiotic treatment. Genome Research, 31(8), 1433–1446. https://doi.org/10.1101/gr.265058.120

Shoemaker, W. R., Chen, D., & Garud, N. R. (2021). Comparative Population Genetics in the Human Gut Microbiome. Genome Biology and Evolution, evab116. https://doi.org/10.1093/gbe/evab116

Wolff, R., Shoemaker, W., & Garud, N. (2023). Ecological Stability Emerges at the Level of Strains in the Human Gut Microbiome. MBio, 0(0), e02502-22. 

Wong, D., & Good, B. (2022). Ecological diversification of rapidly adapting populations. In APS 

March Meeting Abstracts (Vol. 2022, pp. M05-011).

---

## [Decision Letter · Decision Letter 2]

2 Jul 2023

PONE-D-22-30239R2A macroecological perspective on genetic diversity in the human gut microbiomePLOS ONE

Dear Dr. Shoemaker,

Thank you for submitting your manuscript to PLOS ONE. The manuscript is now almost ready for acceptance. However, we have a new reviewer, who has a minor comment, which the author could potentially comment on, and further improve the manuscript.

We look forward to receiving your revised manuscript.

Kind regards,

Karthik Raman, Ph.D.

Academic Editor

PLOS ONE

Journal Requirements:

Reviewers' comments:

Reviewer's Responses to Questions

**Comments to the Author**

1. If the authors have adequately addressed your comments raised in a previous round of review and you feel that this manuscript is now acceptable for publication, you may indicate that here to bypass the “Comments to the Author” section, enter your conflict of interest statement in the “Confidential to Editor” section, and submit your "Accept" recommendation.

Reviewer #3: All comments have been addressed

2. Is the manuscript technically sound, and do the data support the conclusions?

Reviewer #3: Yes

3. Has the statistical analysis been performed appropriately and rigorously? 

Reviewer #3: Yes

4. Have the authors made all data underlying the findings in their manuscript fully available?

Reviewer #3: Yes

5. Is the manuscript presented in an intelligible fashion and written in standard English?

Reviewer #3: Yes

6. Review Comments to the Author

Reviewer #3: The author has addressed all prior reviewer concerns. I have no further concerns, and I commend the author on an interesting piece. One minor point that the author may want to consider is to discuss de novo evolution of coexisting strains from a recent ancestor within the lifespan of a human host. Most of the discussion of ecological dynamics of strains appears to assume rather distant lineages (separated by > thousands of SNPs) colonizing the same host. However, the Zhao et al. reference (ref 1) shows two B. fragilis strains within an individual, separated by a handful of SNPs, where both lineages coexist for a year or more (i.e., stationary dynamics - one lineage does not sweep the other). This would suggest a kind of sympatric speciation (transition from the evolutionary to the ecological regime) is possible over very short timescales.

7. PLOS authors have the option to publish the peer review history of their article (what does this mean?). If published, this will include your full peer review and any attached files.

Reviewer #3: No

---

## [Author Response · Author response to Decision Letter 2]

5 Jul 2023

Reviewer three comment

The author has addressed all prior reviewer concerns. I have no further concerns, and I commend the author on an interesting piece. One minor point that the author may want to consider is to discuss de novo evolution of coexisting strains from a recent ancestor within the lifespan of a human host. Most of the discussion of ecological dynamics of strains appears to assume rather distant lineages (separated by > thousands of SNPs) colonizing the same host. However, the Zhao et al. reference (ref 1) shows two B. fragilis strains within an individual, separated by a handful of SNPs, where both lineages coexist for a year or more (i.e., stationary dynamics - one lineage does not sweep the other). This would suggest a kind of sympatric speciation (transition from the evolutionary to the ecological regime) is possible over very short timescales.

Author response: The reviewer raises an interesting question. They are correct in their assessment that I implicitly assumed that strains are diverged by many SNVs. This is due to the limitations of algorithms such as strain-finder, which uses the distribution of allele frequencies to test for the existence of strain structure within a host, where strain structure cannot be determined if the strains within a host are genetically diverged at only a handful of sites. 

However, strains must ultimately have descended from a common ancestor, and it is worth discussing whether the existence of recently evolved strains can impact the patterns evaluated in this study. A newly evolved strain within a single host is analogous to a species that is only present in a single host. Given that the sampling form of the gamma distribution used in this study succeeds at predicting the prevalence of these species (Grilli, 2020), it should, in principle, be capable of predicting the prevalence of a SNV observed in a single host due to recently evolved strain structure. Across species we find that predictions for low prevalence alleles consistently fail though they tend to succeed for high prevalence alleles (e.g., Figs. S5). It is reasonable to interpret the lack of predictive success for low prevalence alleles as a consequence of said allele being present in a low number of hosts due to evolutionary dynamics, rather than its presence being a reflection of ecological strain structure. However, this interpretation does not mean that recently diverged strains are absent in the cohort of human hosts used in this study. Rather, it is possible that the macroecological lens applied here is insensitive to recently diverged strains that have colonized a low number of hosts similar to the number of hosts that we would expect to find a given allele present at intermediate frequencies due to evolutionary dynamics (e.g., repeated mutation). 

In the revised manuscript I now summarize the above response to address the reviewer’s comment (lines 574-595). 

References

Grilli, J. (2020). Macroecological laws describe variation and diversity in microbial communities. Nature Communications, 11(1), 4743. https://doi.org/10.1038/s41467-020-18529-y

---

## [Editor Report · Decision Letter 3]

7 Jul 2023

A macroecological perspective on genetic diversity in the human gut microbiome

PONE-D-22-30239R3

Dear Dr. Shoemaker,

We’re pleased to inform you that your manuscript has been judged scientifically suitable for publication and will be formally accepted for publication once it meets all outstanding technical requirements.

Kind regards,

Karthik Raman, Ph.D.

Academic Editor

PLOS ONE

---

## [Editor Report · Acceptance letter]

13 Jul 2023

PONE-D-22-30239R3 

A macroecological perspective on genetic diversity in the human gut microbiome 

Dear Dr. Shoemaker:

I'm pleased to inform you that your manuscript has been deemed suitable for publication in PLOS ONE. Congratulations! Your manuscript is now with our production department. 

Kind regards, 

on behalf of

Dr. Karthik Raman 

Academic Editor

PLOS ONE